



**Late Paleocene – early Eocene Arctic Ocean Sea Surface Temperatures;**
**reassessing biomarker paleothermometry at Lomonosov Ridge**
Appy Sluijs[1], Joost Frieling[1], Gordon N. Inglis[2] [*], Klaas G.J. Nierop[1], Francien Peterse[1],
Francesca Sangiorgi[1] and Stefan Schouten[1,3]
[1]Department of Earth Sciences, Faculty of Geosciences, Utrecht University.
Princetonlaan 8a, 3584 CB Utrecht, The Netherlands
[2]Organic Geochemistry Unit, School of Chemistry, School of Earth Sciences,
University of Bristol, Bristol, UK
[3]NIOZ Royal Institute for Sea Research, Department of Microbiology and
Biogeochemistry, and Utrecht University, PO Box 59, 1790AB Den Burg, The
Netherlands
[*] present address: School of Ocean and Earth Science, University of Southampton, UK



**Abstract**
The Integrated Ocean Drilling Program Arctic Coring Expedition on Lomonosov
Ridge, Arctic Ocean (IODP Expedition 302 in 2004) delivered the first Arctic Ocean
sea surface temperature (SST) and land air temperature (LAT) records spanning the
Paleocene-Eocene Thermal Maximum (PETM; ~56 Ma) to Eocene Thermal Maximum
2 (ETM2; ~54 Ma). The distribution of glycerol dialkyl glycerol tetraether (GDGT)
lipids indicated elevated SST (ca. 23 to 27 °C) and LATs (ca. 17 to 25 °C). However,
recent analytical developments have led to: i) improved temperature calibrations and
ii) the discovery of new temperature-sensitive glycerol monoalkyl glycerol tetraethers
(GMGTs). Here, we have analyzed GDGT and GMGT distributions in the same
sediment samples using new analytical procedures, interpret the results following the
currently available proxy constraints and assess the fidelity of new temperature
estimates in our study site.
The influence of several confounding factors on $TEX_{86}$ SST estimates, such as
variations in export depth and input from exogenous sources, are typically negligible.
However, contributions of isoGDGTs from land, which we characterize in detail,
complicate $TEX_{86}$ paleothermometry in the late Paleocene and part of the interval
between the PETM and ETM2. The isoGDGT distribution further supports temperature
as the likely variable controlling $TEX_{86}$ values and we conclude that background early
Eocene SSTs generally exceeding 20 °C, with peak warmth during the PETM (~26 °C)
and ETM2 (~27 °C). We also report high abundances of branched glycerol monoalkyl
glycerol tetraethers throughout (branched GMGTs), most likely dominantly marine in
origin, and show that their distribution is sensitive to environmental parameters. Further
analytical, provenance and environmental work is required to test if and to what extent
temperature may be an important factor.





Published temperature constraints from branched GDGTs and terrestrial vegetation also
support remarkable warmth in the study section and elsewhere in the Arctic basin, with
vegetation proxies indicating coldest month mean temperatures of 6-13 °C. If TEX$_{86}$-
derived SSTs truly represent mean annual SSTs, the seasonal range of Arctic SST was
in the order of 20 °C, higher than any open marine locality in the modern ocean. If SST
estimates are skewed towards the summer season, seasonal ranges were comparable to
those simulated in future ice-free Arctic Ocean scenarios. This uncertainty remains a
fundamental issue, and one that limits our assessment of the performance of fully-
coupled climate models under greenhouse conditions.

**1. Introduction**
The Eocene epoch (56 to 34 million years ago; Ma) has long been characterized by
warm climates. The earliest signs of a balmy Eocene Arctic region – fossil leaves of
numerous plant species – were documented 150 years ago (Heer, 1869). Subsequent
findings identified palms, baobab and mangroves, indicating the growth of temperate
rainforests and year-round frost-free conditions in the Eocene Arctic region
(Schweitzer, 1980; Greenwood and Wing, 1995; Suan et al., 2017; Willard et al., 2019).
Fossils of animals, including varanid lizards, tortoises and alligators also indicate warm
Arctic climates (Dawson et al., 1976; Estes and Hutchinson, 1980). These earliest
findings sparked interest into the climatological mechanisms allowing for such polar
warmth about a century ago (Berry, 1922). Ever since, paleobotanists have focused on
the Arctic plant fossils and have significantly refined their paleoclimatological
interpretation towards estimates of precipitation as well as seasonal and mean annual
temperature (e.g. Uhl et al., 2007; Greenwood et al., 2010; Eberle and Greenwood,
2012; Suan et al., 2017; Willard et al., 2019).





Novel insights in Paleogene Arctic paleoclimate research were made in the years
following the Arctic Coring Expedition 302 (ACEX, Integrated Ocean Drilling
Program (IODP) 2004, Figure 1). This expedition recovered upper Paleocene and lower
Eocene siliciclastic sediments, deposited in a shallow marine environment, in Hole 4A
(87° 52.00 'N; 136° 10.64 'E; 1,288 m water depth), on the Lomonosov Ridge in the
central Arctic Ocean (Backman et al., 2006), deposited at a paleolatitude of ~78 °N,
based on a geological reconstruction (Seton et al., 2012) projected using a
paleomagnetic reference frame (Torsvik et al., 2012) (see paleolatitude.org, Van
Hinsbergen et al., 2015). The sediments are devoid of biogenic calcium carbonate, but
rich in immature organic matter, including terrestrial and marine microfossil
assemblages and molecular fossils that provided a wealth of information regarding
Paleocene and Eocene Arctic climates, environments, and ecosystems (Brinkhuis et al.,
2006; Pagani et al., 2006; Sluijs et al., 2006; Stein et al., 2006; Schouten et al., 2007b;
Stein, 2007; Sangiorgi et al., 2008; Sluijs et al., 2008b; Waddell and Moore, 2008;
Weller and Stein, 2008; Sluijs et al., 2009; Speelman et al., 2009; Speelman et al., 2010;
Barke et al., 2011; Barke et al., 2012; Krishnan et al., 2014; Willard et al., 2019).
As the upper Paleocene and lower Eocene sediments of the ACEX core lack biogenic
calcium carbonate and alkenones, SST reconstructions are based on the biomarker-
based paleothermometer $TEX_{86}$. This proxy is based on membrane lipids (isoprenoid
glycerol dibiphytanyl glycerol tetraethers; isoGDGTs) of Thaumarchaeota, which adapt
the fluidity of their membrane according to the surrounding temperature by increasing
the number of cyclisations at higher temperatures (De Rosa et al., 1980; Wuchter et al.,
2004; Schouten et al., 2013, and references therein). The proxy was introduced in 2002
by Schouten et al. (2002) and was calibrated to mean annual SST using modern marine
surface sediments.



Initial papers suggested that SST increased significantly during two episodes of
transient global warming. Maximum values of ~23°C and ~27 °C occurred during the
Paleocene-Eocene Thermal Maximum (PETM-56 Ma ago, Sluijs et al., 2006) and
Eocene Thermal Maximum 2 (ETM2-54 Ma ago, Sluijs et al., 2009), respectively.
Lower SSTs, generally exceeding 20 °C, characterized the remainder of the early
Eocene (Sluijs et al., 2008b). Such temperatures were immediately recognized to be
remarkably high and could not be explained using fully-coupled climate model
simulations (Sluijs et al., 2006). Even the current-generation of IPCC-class models are
unable to match early Eocene Arctic mean annual SSTs, although reconstructions of
tropical and mid-latitude SSTs and deep ocean temperatures are consistent with some
newer simulations (Frieling et al., 2017; Cramwinckel et al., 2018; Evans et al., 2018;
Zhu et al., 2019).
Since the publication of the ACEX SST records, constraints on the applicability of the
$TEX_{86}$ proxy have tremendously improved (see review by Schouten et al., 2013, and
subsequent work by Taylor, 2013 #1645; Elling et al., 2014; Qin et al., 2014; Elling et
al., 2015; Kim et al., 2015; Qin et al., 2015; Hurley et al., 2016; Zhang et al., 2016).
This work has delivered new constraints on the ecology of Thaumarchaeota, the
dominant depth at which they reside in the ocean and from which depth their isoGDGTs
are exported towards the sea floor. Moreover, it identified potential confounding factors
such as variation in dominant isoGDGT export depth (e.g., Taylor et al., 2013; Kim et
al., 2015), the input of non-Thaumarchaeotal-derived isoGDGTs (e.g., Weijers et al.,
2011; Zhang et al., 2011), growth phase (Elling et al., 2014), and environmental
ammonium and oxygen concentrations (Qin et al., 2015; Hurley et al., 2016), and
several indicators to detect anomalies have been developed. In addition, improvements
in the chromatography method used for GDGT analysis now allow for improved

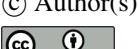



separation of previously co-eluting compounds leading to enhanced analytical precision
and sensitivity (Hopmans et al., 2016). Also, recent work has described new GDGTs
from oceans and sediments, notably glycerol monoalkyl glycerol tetraethers (previously
'H-shaped' GDGTs) (e.g., Schouten et al., 2008; Liu et al., 2012), characterized by a
covalent carbon-carbon bond that links the two alkyl chains, that may be useful for
reconstructing land air temperatures (LAT) (e.g., Naafs et al., 2018a; Baxter et al.,
2019). However, these compounds have not yet been investigated in ancient marine
sediments.
Considering these developments and the paleoclimatological importance of the ACEX
dataset, we re-analyzed the original lipid extracts for the PETM, ETM2 and the interval
spanning these events (Sluijs et al., 2006; Sluijs et al., 2009), according to the latest
chromatography protocols. We also compile published and generate new GDGT data
from modern and Paleogene terrestrial deposits and use these to better assess the
potential confounding influence of isoGDGTs from terrestrial sources, which was
already recognized as a potential problem in the early work (Sluijs et al., 2006).

**2. GDGT-based SST indices, calibration and confounding factors**
*2.1 TEX$_{86}$ and its calibration to SST*
TEX$_{86}$ is based on the relative abundance of 4 different GDGTs (Figure 2), following
(Schouten et al., 2002):
$TEX_{86} = \frac{([GDGT-2]+[GDGT-3]+[Crenarchaeol\ isomer])}{([GDGT-1]+[GDGT-2]+[GDGT-3]+[Crenarchaeol\ isomer])}$    Eq. (1)
where a higher relative abundance of cyclopentane moieties implies higher SSTs.

A number of models are used to calibrate TEX$_{86}$ to SST (Schouten et al., 2002;
Schouten et al., 2003; Schouten et al., 2007a; Kim et al., 2008; Liu et al., 2009; Kim et
al., 2010; O'Brien et al., 2017), all based on a modern ocean surface sediment database.
The currently available culture and mesocosm experiments and surface sediment data
indeed suggest a linear relation, except for polar regions where the $TEX_{86}$ response to
temperature deviates (Kim et al., 2010; Ho et al., 2014; O'Brien et al., 2017). However,
physiological considerations and multiple temperature-dependent GDGT indices might
imply a non-linear relation also at the high temperature end, as can be observed at the
high end of the modern ocean dataset and beyond the reach of the modern ocean
(Cramwinckel et al., 2018). Specifically, at higher temperatures, membrane adaptation
may increasingly be established using isoGDGTs not included in the $TEX_{86}$ ratio
leading to a diminished $TEX_{86}$ response at very high temperatures (Cramwinckel et al.,
2018). A non-linear response has thus been proposed in other calibrations (Liu et al.,
2009; Kim et al., 2010). The most recent non-linear calibration, $TEX_{86}^{H}$ (Kim et al.,
2010), represents an exponential relation between SST and $TEX_{86}$ (Hollis et al., 2019).
Unfortunately, $TEX_{86}^{H}$ is mathematically problematic and has systematic residuals in
the modern ocean (Tierney and Tingley, 2014).
Tierney and Tingley (2014) introduced a spatially-varying Bayesian method to convert
$TEX_{86}$ to SST, which assumes a linear relationship between the two (BAYSPAR). In
deep-time settings, BAYSPAR searches the modern core-top dataset for $TEX_{86}$ values
that are similar to the measured $TEX_{86}$ value within a user-specified tolerance and
draws regression parameters from these modern analogue locations. This approach
yields uncertainty bounds that reflect spatial differences in the slope and intercept terms
and the error variance of the regression model, based on the modern ocean.
Currently, it is generally accepted to present results both using a linear and a non-linear
function (Hollis et al., 2019). The assumption of a linear or non-linear relation between
SST and $TEX_{86}$ leads to very different SST reconstructions for geological samples



yielding $TEX_{86}$ values beyond the modern data set (Kim et al., 2010; Tierney and
Tingley, 2014; Frieling et al., 2017; O'Brien et al., 2017; Cramwinckel et al., 2018).
However, $TEX_{86}$ values of the early Eocene ACEX samples (0.5 – 0.7, Sluijs et al.,
2006; Sluijs et al., 2008b; Sluijs et al., 2009)) are well within the modern ocean
calibration dataset and well above most values observed in the polar regions (Kim et
al., 2010; Tierney and Tingley, 2014; O'Brien et al., 2017), indicating that all
calibrations will yield similar absolute values.

*2.2 Caveats and confounding factors*
Several confounding factors and caveats have been identified that could potentially bias
$TEX_{86}$ data relative to mean annual SST. These notably relate to additions of isoGDGTs
that were not produced in the upper water column by Thaumarchaeota, seasonal biases,
and choices that are made in the calibration between SST and $TEX_{86}$. Below we
summarize methods that have been developed to assess if isoGDGT distributions might
have been biased by confounding factors.

*2.2.1 isoGDGTs of terrestrial origin*
At the time of the first ACEX papers, it was already known that high contributions of
terrestrially-derived isoGDGTs could compromise the $TEX_{86}$ signal (Weijers et al.,
2006). Previous work (Sluijs et al., 2006; Sluijs et al., 2008b; Sluijs et al., 2009) indeed
recognized that high terrestrial contributions of isoGDGTs could be problematic for
portions of the upper Paleocene to lower Eocene interval of the ACEX core based on
high BIT index values. This contribution can be tracked using the Branched and
Isoprenoid Tetraether (BIT) index, a ratio of mostly soil-derived branched GDGTs



(brGDGTs; Figure 2) and Crenarchaeol, which is dominantly marine-derived
(Hopmans et al., 2004; Schouten et al., 2013):
$BIT\ index = \dfrac{([brGDGT-Ia]+[brGDGT-IIa]+[brGDGT-IIIa])}{([brGDGT-Ia]+[brGDGT-IIa]+[brGDGT-IIIa])+[Crenarchaeol])}$    Eq. (2)
Most studies define a BIT value (typically 0.3 or 0.4) above which $TEX_{86}$-derived SST
are unreliable (e.g., Weijers et al., 2006). However, the threshold of 0.4 is conservative
in some settings and the impact of terrigenous GDGTs on reconstructed SST will
depend on the nature and temperature of the source catchment (Inglis et al., 2015). In
addition, a cut-off value based on BIT values is difficult given the relatively large
differences in BIT between labs, which originate from methodological differences
(Schouten et al., 2009). A strong linear relationship between BIT and $TEX_{86}$ values is
often taken as indication of a bias in $TEX_{86}$ through land-derived isoGDGTs to the
marine $TEX_{86}$ signature (e.g., Douglas et al., 2014).

*2.2.2 isoGDGTs of deep water origin*
Thaumarchaeota, the source of most GDGTs in marine waters (Zeng et al., 2019;
Besseling et al., 2020), are ammonium oxidizers (Könneke et al., 2005; Wuchter et al.,
2006a), making them independent of light. Although they occur throughout the water
column, maximum abundances are at depths <200 m, generally around $NO_2$ maxima
(e.g., Karner et al., 2001; Pitcher et al., 2011a). In most oceans, the sedimentary GDGTs
dominantly derive from the upper few hundred meters, based on analyses of suspended
particular organic matter and sediment traps (Wuchter et al., 2005; Wuchter et al.,
2006b; Yamamoto et al., 2012; Richey and Tierney, 2016), although some
contributions from deeper have sometimes been inferred based on [14]C analysis (Shah
et al., 2008). This implies possible contributions of isoGDGTs produced in thermocline
waters. Moreover, contributions of isoGDGTs produced in the deep sea have regionally





been identified (e.g., Kim et al., 2015). Taylor et al. (2013) also found that deeper
dwelling archaea might contribute to the sedimentary isoGDGT assemblage. They
indicate that such deep contributions can be tracked using the GDGT-2/GDGT-3 ratio;
high values of >5 indicate contributions of archaea living deeper in the water column.
Given that upper Paleocene and lower Eocene ACEX sediments were deposited in a
shallow shelf environment (Sluijs et al., 2008b), a significant contribution of deep ocean
archaeal lipids is not expected.

*2.2.3 isoGDGTs of methanotrophic and methanogenic archaea*
Contributions of isoGDGTs to the sedimentary pool might also derive from anaerobic
methanotrophs and/or methanogens. Several indices have been developed to track such
contributions, both based on relatively high contributions of particular isoGDGTs of
these groups of archaea. The Methane Index (MI) was developed to detect the relative
contribution of anaerobic methanotrophic *Euryarchaeota* assumed to be represented by
GDGT-0 but also GDGT-1, 2 and 3 (Zhang et al., 2011) and is therefore defined as
$$MI = \frac{[GDGT-1]+[GDGT-2]+[GDGT-3]}{([GDGT-1]+[GDGT-2]+[GDGT-3]+[Crenarchaeol]+ [Cren.isomer])}$$    Eq. (3)
MI values greater than 0.5 indicate significant anaerobic methanotrophy. Such values
may yield unreliable TEX$_{86}$ values. Another tracer for contributions of anaerobic
methanotrophic archaea is the analogous GDGT-2/Crenarchaeol ratio (Weijers et al.,

237  2011).

Methanogenic archaea can synthesize GDGT-0, as well as smaller quantities of GDGT-
1, GDGT-2 and GDGT-3. The ratio GDGT-0/Crenarchaeol is indicative of
contributions of methanogenic archaea to the isoGDGT pool (Blaga et al., 2009) where
values > 2 indicate substantial contribution of methanogenic archaea. Up to now, high
indices have often been observed near methane seeps or anoxic basins but rarely in open





marine waters. Given the reducing conditions in the sediment and water column at the
study site across the late Paleocene and early Eocene (Sluijs et al., 2006; Stein et al.,
2006; Sluijs et al., 2008b; März et al., 2010), an influence of methane cycling might be
expected.

*2.2.4 isoGDGTs of the 'Red Sea Type'*
Sedimentary isoGDGT distributions from the Red Sea are anomalous to that of other
marine settings by the low abundance of GDGT-0 and the high abundances of the
Crenarchaeol regio-isomer, presumably due to an endemic Thaumarchaeotal
assemblage, and this yields a different relationship between SST and $TEX_{86}$ (Trommer
et al., 2009; Kim et al., 2015). Inglis et al. (2015) attempted to quantify a 'Red Sea-
type' GDGT distribution in geologcal samples using the following index:
$\%GDGTrs = \dfrac{[Crenarchaeol\ isomer]}{([GDGT-0]+[Crenarchaeol\ isomer])} \times 100$    Eq. (4)
However, as noted by Inglis et al., (2015) this ratio is also strongly SST-dependent such
that the Red Sea type GDGT assemblage cannot be discerned from GDGT distributions
that occur at high temperatures in normal open marine settings.

*2.2.5 Seasonal bias*
$TEX_{86}$ is calibrated to mean annual SST. However, particularly in mid and high latitude
areas where production and export production is highly seasonal, the sedimentary
GDGT distribution might not represent annual mean conditions (Wuchter et al., 2006b;
Pitcher et al., 2011b; Mollenhauer et al., 2015; Richey and Tierney, 2016; Park et al.,
2019). This issue should partly be reflected in the calibration uncertainty of the modern
ocean database (several °C, depending on the calibration and method; see section 2.7).
Sluijs et al. (2006; 2008b; 2009) originally argued that the $TEX_{86}$ results from the



ACEX core could be biased towards summer temperature because the export of organic
matter from the surface ocean towards the sediment likely peaked during the season of
highest production, i.e., the summer. However, we also note that the $TEX_{86}$-temperature
relationship is not improved when using seasonal mean ocean temperatures (Kim et al.,
2010; Tierney and Tingley, 2014) and modern observations indicate homogenization
of the seasonal cycle at depth (Wuchter et al., 2006b; Yamamoto et al., 2012; Richey
and Tierney, 2016), implying that seasonality has relatively limited effect on modern
sedimentary $TEX_{86}$ values.

*2.2.6 Additional isoGDGT-based temperature indicators*
The underlying mechanism of $TEX_{86}$ is that isoGDGTs produced at higher SSTs
contain more rings than those produced at low SSTs. Although the combination of
compounds included in $TEX_{86}$ seems to yield the strongest relation with temperature in
the modern ocean (Kim et al., 2010), it implies that isoGDGT ratios other than $TEX_{86}$
also provide insights into SST. One alternative temperature sensitive isoGDGT index
is the Ring Index (RI), which represents the weighed number of cyclopentane rings of
isoGDGTs 0-3, Crenarchaeol and the Crenarchaeol isomer (Zhang et al., 2016), defined
as:
$RI = 0x[\%GDGT - 0] + 1\,x[\%GDGT - 1] + 2\,x[\%GDGT - 2] + 3\,x[\%GDGT - 3] +$
$4\,x\,[\%Crenarchaeol + \%Crenarchaeol\ isomer]$          Eq. (5)
Note that the abundance of GDGT-0 is important for determining the percentage of the
other GDGTs of the total isoGDGT pool.
The close relation between $TEX_{86}$ and RI can also be used to detect aberrant
distributions, including those produced by methanogenic, methanotrophic and
terrestrial sources, as these sources typically contribute disproportionate amounts of



specific lipids. A $RI_{TEX}$, calculated from TEX using the polynomial fit of Zhang et al.
(2016), is subtracted from the RI to arrive at the $\Delta RI$. Cut-off values for sample
deviation from the modern ocean calibration dataset are defined as 95% confidence
limits of the RI-TEX relation, or above $|0.3|$ $\Delta RI$ units.

*2.3 H-shaped branched GDGTs; brGMGTs*
H-shaped branched GDGTs (hereafter referred to as branched glycerol monoalkyl
glycerol tetraethers; brGMGTs; Figure 2) were first identified by Liu et al. (2012) in
marine sediments, who identified a single acyclic tetramethylated brGMGT ($m/z$ 1020).
This compound was later detected within the marine water column and appeared to be
abundant within the oxygen minimum zone (Xie et al., 2014). Naafs et al. (2018a)
identified a larger suite of brGMGTs (including $m/z$ 1048 and 1034), in a quasi-global
compilation of modern peat samples. They argued that these compounds were
preferentially produced at depth, within the anoxic catotelm. Analogous to the
continental paleothermometer based on bacterial brGDGTs produced in surface soils,
termed MBT'$_{5me}$ (Weijers et al., 2011; De Jonge et al., 2014), they showed that the
degree of methylation of brGMGTs in peats relates to mean annual air temperature.
They calculated the degree of methylation of brGDGTs without cyclopentane moieties,
designed for comparison to the methylation of brGMGTs, defined by H-MBT$_{acyclic}$:

$MBTacyclic = \dfrac{brGDGT-Ia}{(brGDGT-Ia+brGDGT-IIa+GDGT-IIa'+brGDGT-IIIa+brGDGT-IIIa')}$ Eq. (6)

$H - MBTacyclic = \dfrac{brGMGT-H1020}{(brGMGT-H1020+brGMGT-H1034+brGMGT-1048)}$     Eq. (7)



Based on the strong relation between MBT*acyclic* and H-MBT*acyclic* in their peat
samples, Naafs et al. (2018a) suggested that the brGMGTs have the same origin as the
brGDGTs, presumably Acidobacteria (Sinninghe Damsté et al., 2011; Sinninghe
Damsté et al., 2018). In addition, they showed that the abundance of brGMGTs relative
to the total amount of brGMGTs and brGDGTs positively correlates with mean annual
air temperature, suggesting that the covalent bond in the brGMGTs is used to maintain
membrane stability at higher temperature (Naafs et al., 2018a).
Baxter et al., (2019) identified a total of seven different brGMGTs in the mass
chromatograms with *m/z* 1020, 1034 and 1048 from a suite of African lake sediments
(Figure 2), and found their relative distribution to correlate to mean annual air
temperature. Accordingly, they proposed a proxy for mean annual air temperature
termed brGMGT-I (see Figure 2 for the molecular structures referred to here):
$$brGMGT - I = \frac{[H1020c] + [H1034a] + [H1034c]}{[H1020b] + [H1020c] + [H1034a] + [H1034c] + [H1048]} \qquad \text{Eq. (8)}$$

**3. Material and Methods**
We used the polar fractions previously analyzed by Sluijs et al. (2006; 2009) from the
PETM through ETM2 interval at IODP Expedition 302 Hole 4A. These fractions
originate from a total lipid extract produced using a Dionex Accelerated Solvent
Extractor and fraction separations by Al$_2$O$_3$ column chromatography using
hexane:dichloromethane (DCM) (9:1, volume:volume) and DCM:methanol (1:1) to
yield the apolar and polar fractions, respectively. Polar fractions were re-dissolved in
hexane:isopropanol (99:1) and passed through a 0.45-μm polytetrafluoroethylene filter.
This fraction was then analyzed by high-performance liquid chromatography (HPLC)
and atmospheric pressure chemical ionization–mass spectrometry using an Agilent
1260 Infinity series HPLC system coupled to an Agilent 6130 single-quadrupole mass



spectrometer at Utrecht University following Hopmans et al. (2016) to measure the
abundance of GDGTs. Based on long-term observation of the in-house standard, the
analytical precision for TEX$_{86}$ is ±0.3 °C.
To gain further insights into the potential impact of terrestrial isoGDGT input on TEX$_{86}$
values, we compiled isoGDGT and brGDGTs distributions from modern peats (n = 473,
Naafs et al., 2017) and early Paleogene lignites (n = 58, Naafs et al., 2018b). Note, the
fractional abundance of Crenarchaeol isomer was not reported in the early Paleogene
dataset of Naafs et al. (2018b). We therefore re-analyzed the polar fractions of their
early Paleogene lignite extracts via HPLC-MS using a ThermoFisher Scientific Accela
Quantum Access at the University of Bristol following Hopmans et al. (2016). Based
on long-term observation of the in-house standard, the analytical precision for TEX$_{86}$
is ±0.3 °C for both labs.

**4. Results**
The new GDGT distributions (Supplementary Table) are consistent with the TEX$_{86}$ and
BIT index data generated over a decade ago using the old analytical HPLC setup
(Hopmans et al., 2000; Hopmans et al., 2016) (Figure 3). TEX$_{86}$ exhibits some scatter
but the slope of the regression is 0.98 for the entire dataset, which is indistinguishable
from the 1:1 line. The scatter is minor compared to the uncertainties inherent to
calibrations that transfer these values to SST. Less scatter is apparent in the BIT record
but the original BIT index values were slightly higher at the higher end recorded here
(~0.5), indicated by a shallower slope of the regression (0.92), consistent with previous
analyses with the new analytical setup (Hopmans et al., 2016). This does not impact
previous qualitative interpretations of this record (Sluijs et al., 2006; Sluijs et al., 2008b;
Sluijs et al., 2009). In the discussion section, we assess indicators of potential
confounding factors (section 2.2), including the influx of terrestrially-derived
isoGDGTs to the sediments (Figures 4, 5 and S1) and several indices related to methane
and depth of production (Figures 6).
Although we did not detect significant amounts of isoprenoid GMGTs, high
abundances of various brGMGTs, in total between 10 and 45% of the total brGDGT
assemblage (Figure 7), are present in the ACEX samples. Specifically, we can
consistently identify at least 5 peaks across the mass chromatograms of $m/z$ 1020, 1034
and 1048. Based on their (relative) retention times and overall distribution we were able
to apply the nomenclature of Baxter et al. (2019) to 5 of these and assign individual
peaks to previously identified compounds (Figure S2). Abundances of brGMGTs
relative to brGDGTs increase during the PETM. Furthermore, the proposed temperature
indicators based on brGMGTs show mixed results, with some showing a clear response
to the PETM (Figure 7d) while others do not (Figure 7e).

**5. Discussion**
*5.1 IsoGDGT provenance*
*5.1.1 Contributions of soil-derived isoGDGTs*
As noted by Sluijs et al. (2006), Paleocene samples yield anomalously high abundances
of GDGT-3, likely derived from a terrestrial source. To assess the temperature change
during the PETM, they therefore explored a $TEX_{86}$ calibration without this moiety,
termed $TEX'_{86}$. However, $TEX'_{86}$ has not been widely used outside the Paleogene Arctic
because the anomalous abundances of GDGT-3 have not been recorded elsewhere. In
addition, high contributions of GDGT-3 from terrestrial input would also be associated
with an increase in the abundance of other isoGDGTs. We therefore consider the late
Paleocene temperature estimates unreliable. Indeed, recent $TEX_{86}$-based global SST





compilations and comparison to climate simulations for the PETM excluded the ACEX
record because the TEX$_{86}$' calibration complicates the comparison to other regions
(Frieling et al., 2017; Hollis et al., 2019) and has not been applied elsewhere.
Input of soil organic matter is consistent with Willard et al. (2019) who established that
the brGDGT assemblage is dominantly soil derived as opposed to being produced in
the coastal marine environment (Sinninghe Damsté, 2016). This observation is based
upon the weighted average number of rings in the tetramethylated brGDGTs (#rings$_{tetra}$)
which generally does not exceed 0.4 to 0.7 in the global soil calibration dataset
(Sinninghe Damsté, 2016). In the ACEX record, #rings$_{tetra}$ is always below 0.21
(Willard et al., 2019), consistent with a dominant soil source. This indicates that
brGDGT abundances, brGDGT distributions and the BIT index are reliable indicators
of the relative supply of terrestrially-derived isoGDGTs into the marine basin.
The Paleocene section of the dataset also stands out regarding its relation between BIT
index and TEX$_{86}$ (Figure 4), which confirms its anomalous nature. During the PETM,
TEX$_{86}$ values are higher due to warming and BIT values lower, which was attributed to
sea level rise during the hyperthermals resulting in a more distal position relative to the
terrestrial GDGT source (Sluijs et al., 2006; Sluijs et al., 2008a). From the remainder
of the dataset, the interval between 371.0 and 368.0 mcd, just below ETM2, stands out.
This interval was previously recognized by Sluijs et al. (2009) to reflect the most open
marine environment in the studied section, with dominant marine palynomorphs and
biomarkers. They also found that high BIT values correspond to low TEX$_{86}$ values
within that interval and therefore they implemented a subjective cut-off BIT value of
0.3, above which TEX$_{86}$-derived SSTs were considered unreliable. Although the
relation between BIT and TEX$_{86}$ exhibits much scatter, the new analyses supports the
notion that higher influx of terrestrial isoGDGTs lowers TEX$_{86}$ values, explaining 26%



of the variation in a linear regression (Figure 4). The nature of this influence is
determined by the relative abundance of terrestrial isoGDGTs and their $TEX_{86}$ value.
The $TEX_{86}$ value at the terrestrial endmember of BIT = 1, assuming various types of
relations, centers around 0.5. The remainder of the data does not show a clear relation
between BIT and $TEX_{86}$ although some of the lowest $TEX_{86}$ values correspond to high
BIT values, suggesting that the terrestrial endmember contributed isoGDGT
assemblages with relatively low $TEX_{86}$ values in other intervals as well.
Interestingly, the relatively low degree of cyclization of soil-derived isoGDGTs in the
early Eocene contrasts starkly with the anomalous contributions of GDGT-3 in the
Paleocene. This implies that the distribution of supplied terrestrial isoGDGTs differed
strongly between the Paleocene and Eocene part of the studied section.
The impact of soil-derived isoGDGTs also emerges from the Ring Index approach of
Zhang et al. (2016, see section 2.6). The difference between the Ring Index and $TEX_{86}$
at the onset of the PETM is mainly controlled by Crenarchaeol, which is comparatively
low in abundance in the Paleocene but highly abundant in the PETM. This increase is
likely associated with sea level rise during the PETM because Crenarchaeol is
predominantly produced in the marine realm. It is also consistent with a drop in BIT
index values and the relative abundance of terrestrial palynomorphs (Sluijs et al.,
2008a). The approach of Zhang et al. (2016) also confirms that many isoGDGT
distributions exhibit an anomalous relation between $TEX_{86}$ and the Ring Index relative
to the modern core top dataset, with ΔRI values >0.3 (Figure 6). Importantly, all
samples with ΔRI values >0.3 have BIT values above 0.35, indicating that contributions
of soil-derived iso-GDGTs dominate non-temperature effects in the distributions. We
therefore discard $TEX_{86}$-derived SSTs for samples with BIT values >0.35.



We attempt to further constrain the potential contribution of terrestrially-derived
isoGDGTs by determining the abundance of isoGDGTs relative to brGDGTs in modern
peat samples (Naafs et al., 2017) and early Paleogene lignites (Naafs et al., 2018b, the
isoGDGT data are published here). The absolute concentrations of brGDGTs in the
ACEX samples are then used to estimate the potential contribution of terrestrially-
derived isoGDGTs to the samples. To this end, we use the fractional abundance of the
various isoGDGTs in available global terrestrial sediment calibration datasets,
specifically modern peats and Paleogene lignites (Figure 5). Then, we estimate the
abundance of these terrestrially-derived isoGDGTs in our ACEX samples by scaling
this fraction to the measured abundances of brGDGTs and isoGDGTs in our ACEX
samples, following
$Terrestrial\ fraction\ of\ isoGDGT$ n =
$(Fraction\ of\ isoGDGT\text{n}\ in\ terrestrial\ test\ dataset * \frac{sum(brGDGTs))}{abundance\ of\ isoGDGT\ n})$ Eq. (9)
where $n$ represents the specific analyzed GDGT.
This leads to estimates of the potential relative contributions of the individual
isoGDGTs derived from land in the ACEX samples based on the entire modern peat
dataset (Naafs et al., 2017), modern peats from regions with MAT exceeding 15°C
(Naafs et al., 2017) and Paleogene lignites (Naafs et al., 2018b, this paper, Figures 5
and S1). This shows that Crenarchaeol and Crenarchaeol-isomer remain almost
exclusively marine even with high brGDGT concentrations. However, we show that
GDGT-1, GDGT-2 and GDGT-3 all have potentially large terrestrial contributions in
the ACEX samples (Figure 5), more concentrated in specific stratigraphic intervals
(Figure S1). In the most extreme cases, the modeled contributions of terrestrial
isoGDGTs, based on the measured brGDGTs and modern peat dataset is higher than
the actually measured isoGDGT abundances (terrestrial fraction higher than 1). This is





principally seen in iGDGT-2 and 3, and predominantly when we calculate the isoGDGT
contribution using the Paleogene lignite database. This particular assumption thus
clearly leads to overestimates of the amount of terrestrially sourced isoGDGTs in our
setting. However, the trends between the modern peats, warm modern peats and
Paleogene lignites are essentially identical and give some indication which isoGDGTs
are most likely to be affected and across which intervals.
Interestingly, particularly GDGT-3 is shown to be affected in ACEX samples if the
terrestrial contribution of isoGDGTs is analogous in distribution to that of warm
modern peats and/or Paleogene lignites (Figure 5), which qualitatively matches the
distributions in the ACEX samples. This is principally because GDGT-3 is the least
abundant marine isoGDGT included in our analyses, whereas it is often as abundant as
GDGT-1 and 2 in terrestrial settings (Fig. 5).

*5.1.2 Contributions of methanotrophic or methanogenic archaea?*
The depositional environment at the study site, with ample (export) production,
sediment organic matter content, and low oxygen conditions at the sediment-water
interface (Sluijs et al., 2006; Stein et al., 2006; Stein, 2007; Sluijs et al., 2008b; Sluijs
et al., 2009; März et al., 2010), may have been suitable for abundant methanogenic and
methanotrophic archaea, potentially contributing to the sedimentary isoGDGT
assemblage. However, our GDGT-2/Crenarchaeol values (<0.23; Figure 6) are far
below values that suggest significant isoGDGT contributions of methanotrophic
*Euryarchaeota* as described by Weijers et al. (2011). Also MI values (maximum
observed 0.31) are generally below proposed cut off values (0.3-0.5, Zhang et al., 2011)
that suggest such contributions. Finally, GDGT-0/Crenarchaeol ratios (<1.4) remain



below the cut-off value of 2 throughout the section (Figure 6), also making a significant
isoGDGT contribution from methanogens highly unlikely (Blaga et al., 2009).

*5.1.3 Contributions of deep-dwelling archaea?*
Taylor et al. (2013) showed that GDGT-2/GDGT-3 ratios correspond to depth of
production, with high values (>5) where water depth is >1000 m. We record values
between 1 and 4 from the bottom of the study section up to ~371.2 mcd (Figure 6),
which supports a dominant production in the surface ocean based on the modern
calibration data set (Taylor et al., 2013). However, the overlying interval up to ~368.3
mcd has much higher values and also highly variable values averaging 7.4 and with
peak values of 10-14. Such values suggest significant contributions of isoGDGTs
produced at water depths of several kilometers according to the analyses by Taylor et
al. (2013).
However, all paleoenvironmental information generated based on the sediments as well
as tectonic reconstructions of Lomonosov Ridge – a strip of continental crust that
disconnected from the Siberian margin in the Paleocene - has indicated a neritic setting
of the drill site at least up to the middle Eocene (e.g., O'Regan et al., 2008; Sangiorgi
et al., 2008; Sluijs et al., 2008a; Sluijs et al., 2009). Although a drop in BIT index and
a change in the palynological assemblages support a change towards a somewhat more
distal position relative to the shoreline at ~371.2 mcd, the sediment remains dominantly
siliciclastic and organic terrestrial components, particularly pollen and spores, remain
abundant (Sluijs et al., 2008a; Sluijs et al., 2008b). The high GDGT-2/GDGT-3 ratio
values can therefore not be explained by contributions of deep dwelling archaea.
Indeed, increased contributions of isoGDGTs produced at depth would be expected to
have caused a systematic cold bias. However, based on linear regression analysis the





large variability in GDGT-2/GDGT-3 ratios is unrelated to the recorded variability in
$TEX_{86}$ values.
Intriguingly, in a study of the last 160 kyr in the South China Sea, Dong et al. (2019)
found that very high GDGT-2/GDGT-3 ratios (~9 but up to 13) correspond with high
values in nitrogen isotope ratios, interpreted to reflect low contributions in diazotroph
$N_2$ fixation and enhanced upwelling. In our records, the high GDGT-2/GDGT-3 ratios
are associated with normal marine conditions and the dinocyst assemblages are not
indicative of upwelling conditions (Sluijs et al., 2009). Unfortunately, the available
nitrogen isotope record (Knies et al., 2008) does not cover our study interval in
sufficient resolution to assess a relation with diazotroph activity. As such, the cause of
the high GDGT-2/GDGT-3 ratios in this interval remains unclear but we consider it
highly unlikely to relate to contributions of very deep dwelling Thaumarchaeota.

*5.1.4 Oxygen concentrations and ammonium oxidation rates*
A variety of non-thermal factors can impact $TEX_{86}$ values, including ammonium and
oxygen concentrations and growth phase (Elling et al., 2014; Qin et al., 2014; Hurley
et al., 2016). Across the studied interval of the ACEX core, several intervals of seafloor
and water column anoxia have been identified based on organic and inorganic proxies,
notably during the PETM and ETM2 (Sluijs et al., 2006; Stein et al., 2006; Sluijs et al.,
2008b; Sluijs et al., 2009; März et al., 2010).
Particularly suspect is an interval of low $TEX_{86}$ values that marks the middle of the
ETM2 interval, directly following a ~4 °C warming at its onset (Sluijs et al., 2009).
This interval is also marked by the presence of sulfur-bound isorenieratane (Sluijs et
al., 2009), a derivative of isorenieratene, a biomarker produced by the brown strain of
green sulfur bacteria that require light for photosynthesis and free sulfide, indicating



euxinic conditions in the (lower) photic zone (Sinninghe Damsté et al., 1993). We also
record a concomitant shift in several methane-related indicators, GDGT-2/GDGT-3
ratio values and the ΔRI. A mid-ETM2 cooling signal has not been recorded at other
study sites and this interval marks the occurrence of pollen of thermophilic plants such
as palms and baobab (Sluijs et al., 2009; Willard et al., 2019). Therefore, the low $TEX_{86}$
values were suggested to reflect thaumarcheotal depth migration to the deeper
chemocline due to euxinic conditions (Sluijs et al., 2009), similar to the modern Black
Sea (Coolen et al., 2007; Wakeham et al., 2007) and the Mediterranean Sea during
sapropel formation (Menzel et al., 2006).
More recent work has indicated that the isolated marine Thaumarchaeotal species
*Nitrosopumilus maritimus* produces lower $TEX_{86}$ values with higher ammonia
oxidation rates (Hurley et al., 2016) and $O_2$ concentrations (Qin et al., 2015). Although
this observation is difficult to extrapolate to the total response of the Thaumarcheotal
community in the marine environment on geological time scales, lower $O_2$ availability
should lower oxidation rates leading to higher $TEX_{86}$ values (Qin et al., 2015; Hurley
et al., 2016). However, we record a drop in $TEX_{86}$ values with the development of
anoxia during ETM2. The nature of the anomalously low cyclization in the ETM2
isoGDGT assemblage, which pass all quality tests regarding GDGT distribution (Figure
6), remains therefore elusive. In general, however, if the relatively restricted and low-
$O_2$ setting had any impact on $TEX_{86}$ values, these culture studies (Qin et al., 2015;
Hurley et al., 2016) suggest it would have led to an underestimate of the SST.

*5.2 Origin and environmental forcing of brGMGTs*
The relative abundances of brGMGTs in our samples are surprisingly high. On average,
they comprise 25% of the total branched GDGT and GMGT assemblage. The limited





literature on modern occurrences implies that both terrestrial and marine sources may
have contributed to the brGMGT assemblage. Data from the marine sediments (Liu et
al., 2012) and the water column (Xie et al., 2014), clearly shows production within the
marine realm. Their occurrence in modern peats (Naafs et al., 2018a), lake sediments
(Baxter et al., 2019) and Paleogene lignites (Inglis et al., 2019) might also imply
transport from land to marine sediments. A soil-derived source is currently
unsupported, as they were most often below detection limit in recent studies of
geothermally heated soils (De Jonge et al., 2019) and a soil transect from the Peruvian
Andes (Kirkels et al., 2020). The brGMGT abundances we record are close to the
maximum abundance found in modern peats (Naafs et al., 2018a). However, significant
input of peat-derived organic matter into our study site is inconsistent with the low input
of peat-derived *Sphagnum* spores (Willard et al., 2019). Alternatively, the high
abundance of brGMGTs could also be related to subsurface production, which was
invoked by Naafs et al. (2018a) to explain very high abundance of brGMGTs in an early
Paleogene lignite. Collectively, however, we argue that production in the marine realm
may be an important contributor to the brGMGT pool in our setting.
Several factors may contribute to the rise in the abundance of brGMGTs relative to
brGDGTs across the PETM. Higher relative abundances of brGMGTs in modern peats
generally occur at higher mean annual air temperatures (Naafs et al., 2018a) and so this
signal could relate to warming during the PETM if their origin at the study site is
terrestrial. However, we consider it likely that a large part of the brGMGTs assemblage
is of marine origin. If so, the rise in brGMGT abundance likely relates to the previously
recorded (Sluijs et al., 2006; Sluijs et al., 2008b) sea level rise during the PETM at the
study site, causing an increase in marine brGMGT production relative to terrestrial
brGDGT supply to the study site. This is consistent with the inverse correlation between



brGMGT abundance and the BIT index (Figure 7). Lastly, if the production of marine
brGMGTs was focused in oxygen minimum zones (Xie et al., 2014), the development
of low oxygen conditions in the water column based on several indicators, such as the
presence of isorenieratane (Sluijs et al., 2006), might have increased the production of
brGMGTs in the water column. It is also possible that all of these factors contributed
to the changes in abundance of brGMGTs relative to brGDGTs across the PETM.
The brGMGT-I proxy does not produce temperature trends similar to those seen in
TEX$_{86}$ or MBT'$_{5me}$ (Figure 7e). If the majority of the brGMGTs is of marine origin, this
indicates that brGMGTs produced in the marine realm do not respond to temperature
as was hypothesized based on the African Lake dataset by Baxter et al. (2019).
Also the application of the H-MBT*acyclic* index (equation 7) appeared problematic
because, similar to Baxter et al. (2019), we identified several more different isomers
than Naafs et al. (2018a, who developed this index) detected in their peat samples. It
therefore remains unclear which of our peaks should be used to calculate the H-
MBT*acyclic* index values. We therefore show the two plausible options. For the first,
we use all peaks with *m/z* 1020, 1034 and 1048 (HMBT all in Figure 7) within the
expected retention time window. However, based on our chromatography, we consider
it more likely that the dominant peaks identified by Naafs et al. (2018a) at *m/z* 1020 and
1034 represent H1020c and H1034b, respectively, and therefore use only those in
addition to the single identifiable peak at *m/z* 1048 as a second option. Both options
show a clear rise across the PETM, although the HMBT (H1020c, H1034a) shows a
larger signal and somewhat better correspondence in absolute values to MBT*acyclic*,
though with more scatter. A close correspondence between MBT*acyclic* and HMBT
was also found in a recent analysis of a lignite that seems to correspond to the PETM,





although interestingly, no apparent relation with temperature was found (Inglis et al.,

616    2019).

If a pronounced part of brGMGTs within the terrestrially-dominated Paleocene part of
the section is of terrestrial origin, it is possible that the drop in the relative contribution
of terrestrially-derived versus marine brGMGTs influenced these records. However, if
the dominant source of the brGMGTs was marine throughout the record, the increase
in methylation possibly relates to warming. This would not be unprecedented as marine-
produced brGDGTs show an increase in methylation as a function of temperature
(Dearing Crampton-Flood et al., 2018), and also isoprenoid GMGTs produced in
marine sediments by archaea (below detection in our samples) incorporate additional
methyl groups at higher sediment temperatures (Sollich et al., 2017). However, along
with the unresolved brGMGT sourcing, during the PETM at the study site also water
column oxygen concentrations and pH changed, which potentially affected
distributions. Extensive evaluation of brGMGT distributions in modern samples is
therefore required to assess the proxy potential.

*5.3 Uncertainty on $TEX_{86}$-based SST estimates.*
*5.3.1 Uncertainty based on calibration dataset*
To calculate SSTs, we use the BAYSPAR method (Tierney and Tingley, 2014) – which
assumes a linear relationship between $TEX_{86}$ and SST -   and $TEX_{86}^{H}$ (Kim et al., 2010)
– which assumes a non-linear relationship between $TEX_{86}$ and SST. Differences
between these calibrations are smaller than the calibration errors (Figure 6) because the
$TEX_{86}$ values in the ACEX dataset all fall well within the range of the modern core top
calibration. Collectively, taken at face value, the data imply that mean annual SSTs



varied between 18 °C and 28 °C in the early Eocene, providing strong evidence for
remarkable early Eocene warmth in the Arctic region.
The $TEX_{86}^{H}$ calibration implies a calibration error of 2.5 °C (residual mean standard
error; RSME) (Kim et al., 2010). The BAYSPAR method yields possible values that
range ~6 °C from the most probable value (Figure 6), but these uncertainty estimates
are more comparable than is immediately apparent as this analysis takes a 90%
confidence interval compared to the 68% probability of RSME. However, all of the
calibrations and methods to obtain values and uncertainties are based on a modern core-
top dataset and thus implicitly include potential confounding factors such as seasonality
and depth of production and export. However, there is no (quantitative) constraint on
any of these parameters in the calibration data set. This is particularly important for the
studied region because it represents a polar endmember of the marine environment with
highly seasonal production and export and potentially high seasonality in temperature.
In the modern ocean, relations between SST and $TEX_{86}$ in the Arctic and ice-proximal
Southern Ocean settings differ from the global ocean, attributed to a change in
viscoelastic adaptation to temperature at the low end and/or a change in the
Thaumarchaeotal community (Kim et al., 2010; Ho et al., 2014; Tierney and Tingley,
2014). This may mask potential confounding factors that may be relevant specifically
to polar environments. This is important for our case, a situation in which the polar
regions were ice free and the functioning of physical, chemical and biological ocean
systems were fundamentally different from present day. This implies that any
uncertainty calculated based on the modern database, regardless whether it is done
based on traditional regression analyses or BAYSPAR, has no direct value for
determination of uncertainty in our case because the caveats and confounding factors





do not influence uncertainty in the same way in the Eocene as in the modern.
Quantification of uncertainty is at this point, therefore, extremely difficult.

*5.3.2 Constraints from independent proxy data*
Independent proxy data may provide additional constraints. The appearance of the
dinoflagellate cyst genus *Apectodinium* during the PETM and ETM2 in the Arctic basin
(Sluijs et al., 2006; Sluijs et al., 2009; Harding et al., 2011) provide qualitative support
for pronounced warming and apparent subtropical conditions. Recent efforts to quantify
the paleoecological affinities of this now extinct genus have suggested a required
minimum temperature of ~20°C (Frieling et al., 2014; Frieling and Sluijs, 2018).
Although this value is partly based on $TEX_{86}$ data from the ACEX cores, it is supported
by data from an epicontinental site in Siberia (Frieling et al., 2014).
A second line of independent proxy evidence includes vegetation reconstructions. As
indicated above, the $TEX_{86}$ results are qualitatively consistent with the ample evidence
for thermophilic plants and animals in the Arctic (e.g., Heer, 1869; Schweitzer, 1980;
Greenwood and Wing, 1995; Uhl et al., 2007; Suan et al., 2017). Particularly valuable
are minimum winter temperature tolerances for specific plant species. Palynological
analyses have indicated the presence of palm and baobab pollen within the PETM and
ETM2 intervals in the ACEX cores (Sluijs et al., 2009; Willard et al., 2019). Modern
palms are unable to tolerate sustained intervals of frost whilst sexual reproduction is
limited to regions where the coldest month mean temperature is significantly above
freezing (Van der Burgh, 1984; Greenwood and Wing, 1995). The latter was recently
quantified to be $\geq 5.2$ °C (Reichgelt et al., 2018). The presence of baobab within the
PETM interval and ETM2 support mean winter air temperatures of at least 6 °C
(Willard et al., 2019). Importantly, these plants were not encountered in the intervals



outside the PETM and ETM2, suggesting background coldest month mean air
temperatures were potentially too low (<6ºC) to support these megathermal vegetation
elements.
Pollen of palms and *Avicennia* mangroves were recently identified in time-equivalent
sections in Arctic Siberia (Suan et al., 2017). Although the details of stratigraphic
context of these records may be somewhat problematic, these findings provide good
evidence for very high coldest month mean temperatures, both air (>5.5 °C) and SST
(>13 °C) during the late Paleocene and early Eocene (Suan et al., 2017).
Apparently conflicting evidence comes from the occurrence of glendonites and erratics
in specific stratigraphic levels in Paleocene and Eocene strata in Spitsbergen,
interpreted to reflect cold snaps in climate (Spielhagen and Tripati, 2009). Some of
these stratigraphic levels are very close to (or even potentially within) the PETM,
considering the local stratigraphic level of the PETM (Cui et al., 2011; Harding et al.,
2011), although glendonites and erratics have not been found at the exact same
stratigraphic levels as thermophilic biota (Spielhagen and Tripati, 2009). The formation
and stability of ikaite (the precursor mineral of the diagenetic glendonites) in
Spitsbergen was dependent on relatively low temperature, arguably persistent near-
freezing sea water temperatures in the sediment (Spielhagen and Tripati, 2009).
However, glendonite occurrences, some also in Mesozoic sediments in mid-latitude
regions, have recently also been linked to methane seeps and so the specific temperature
constraints implied by glendonites under such conditions are subject of debate (e.g.,
Teichert and Luppold, 2013; Morales et al., 2017). Clearly, however, the glendonite
occurrences may imply episodes of colder climates and follow up work should apply
temperature reconstructions based on biomarkers or biota on corresponding strata to
assess proxy consistency.



This estimate on seasonal minima provides an important constraint on Arctic
climatology during the PETM and ETM2. Most likely, the palms and baobabs grew
close to the shore, where the relative heat of the ocean kept atmospheric temperatures
relatively high during the winter. If minimum winter SSTs were in the range of the SST
reconstructions based on the nearby *Avicennia* mangrove pollen (Suan et al., 2017),
which for open ocean settings would perhaps amount to ~10 °C, then summer SST must
have soared to at least 30 °C in summer if $TEX_{86}$–based SST reconstructions of ~20 °C
truly reflects the annual mean. It would imply an SST seasonality of ~20 °C, much
higher than any modern open marine setting, let alone the Arctic Ocean. In the present
day Arctic Ocean, heat is seasonally stored and released in sea ice melting and freezing,
and sea ice cover insulates the ocean and reflects much sunlight, resulting in a seasonal
cycle of not more than 1.5 °C, even in ice-free regions (Chepurin and Carton, 2012).
However, coupled model simulations have indicated that the future loss of sea ice will
greatly enhance the seasonal SST range to up to 10 °C in 2300 given unabated $CO_2$
emissions (Carton et al., 2015). With year-round snow and ice-free conditions, even
stronger summer stratification during the Eocene due to higher greenhouse gas
concentrations and fresh-water supply through an enhanced hydrological cycle
(Pierrehumbert, 2002), a near-shore 20 °C seasonal cycle in Arctic Ocean SST may not
be unrealistic, although it remains inconsistent with current-generation fully coupled,
relatively low resolution, model simulations (e.g., Frieling et al., 2017).
Constraints from the total pollen assemblages in the ACEX cores based on a nearest
living relative approach suggest Arctic mean annual temperatures on land of 13-18 °C,
and summer temperatures significantly exceeding 20 °C during the PETM and ETM2
(Willard et al., 2019). Although these estimates come with much larger uncertainty than
winter temperatures and may suffer from the non-analogous setting, they are generally





lower than our $TEX_{86}$ values. Also the brGDGT-based paleothermometer MBT'$_{5me}$ (De
Jonge et al., 2014) indicates lower temperatures mean annual air temperatures than
reported from $TEX_{86}$ (Willard et al., 2019, Figure 7). These data, derived from the same
UHPLC/MS analyses as the isoGDGT data presented here, indicate mean annual air
temperatures averaging ~18 °C during the PETM, with a residual mean calibration error
of 4.8 °C. This value is ~7 °C lower than earlier estimates based on a slightly different
method, analytical procedure and a smaller modern calibration dataset (Weijers et al.,

745  2007).


*5.4 State of constraints on Paleocene-Eocene Arctic temperatures*
To unlock the unique premise of Eocene climates for testing the skill of current-
generation fully coupled climate models under high greenhouse gas forcing, proxy data
and models are ideally approached separately. Among the most important implications
of the Arctic temperature estimates are reconstructions of the meridional temperature
gradients. Importantly, not a single simulation using an IPCC-class model of early
Paleogene climate has produced Arctic annual mean sea surface temperatures close to
the ACEX $TEX_{86}$-based reconstructions without unrealistically high tropical SSTs
(Lunt et al., 2012). Recent simulations using the Community Earth System Model 1
(CESM-1) using Eocene boundary conditions produced climates that correspond to
SST reconstructions in many ocean regions based on several proxies, but still produced
cooler mean annual SSTs for the Arctic Ocean than suggested by $TEX_{86}$ (Frieling et al.,
2017; Cramwinckel et al., 2018; Zhu et al., 2019). $TEX_{86}$ also indicates SSTs higher
than in these model simulations at several sites along the Antarctic margin (Bijl et al.,
2009; Bijl et al., 2013). The question thus remains if the conversion of $TEX_{86}$ values
towards mean annual SST using any modern core-top calibration for high latitude



Paleogene locations is valid, or if the climate models still significantly underestimate
polar temperatures. Certainly, if interpreted as mean annual SST, $TEX_{86}$-based
estimates are high compared to the few available additional estimates, notably based on
vegetation, but the latter also suffer from similar uncertainties.
A few biases might exaggerate meridional temperature gradients as indicated from
$TEX_{86}$. First, the flat Eocene temperature gradient implied by $TEX_{86}$ was suggested to
result from erroneously calibrating the proxy to SST rather than to the temperature of
the subsurface (Ho and Laepple, 2016). The rationale is that the meridional temperature
gradient is smaller in deeper waters than it is in the surface. However, the idea was
contested for multiple reasons, including the fact that sediments at most Eocene study
sites, such as the ACEX site, were deposited at a depth of less than 200 m, making the
application of a deep subsurface (>1000m) calibration inappropriate (Tierney et al.,
2017). Moreover, recent analyses have indicated that the $TEX_{86}$ signal dominantly
reflects temperature of top 200 m of the water column (Zhang and Liu, 2018).
Secondly, as suggested previously (Sluijs et al., 2006), if $TEX_{86}$ were biased towards
any season in the non-analogue Arctic Ocean, it would be the summer, the dominant
season of organic matter export towards the seafloor through fecal pelleting or marine
snow aggregates. Vegetation suggests very high winter continental coldest month mean
air temperatures of at least 6-8 °C (Sluijs et al., 2009; Suan et al., 2017; Willard et al.,
2019), coastal coldest month mean SSTs of >13 °C (Suan et al., 2017), and terrestrial
mean annual and warmest month mean temperature on land of 13-21 °C and >20°C,
respectively (Suan et al., 2017; Willard et al., 2019) (see section 5.3.2). These estimates
are closer to the most recent model simulations and lower than the existing $TEX_{86}$ (e.g.,
Zhu et al., 2007; Frieling et al., 2017). If $TEX_{86}$-implied SST of ~25 °C is skewed
towards a summer estimate, this would decrease the model-data bias regarding the



meridional temperature gradient estimates. Given the current uncertainties in the use of
TEX$_{86}$ for the non-analogue Arctic Ocean, we however cannot independently constrain
this.

**6. Conclusions**
We analyzed isoGDGT and brGMGT (H-shaped branched GDGT) distributions in
sediments recovered from the Paleocene-Eocene Thermal Maximum (PETM; ~56 Ma)
to Eocene Thermal Maximum 2 (ETM2; ~54 Ma) interval on Lomonosov Ridge, Arctic
Ocean using state-of-the-art analytical procedures, compare them to the original dataset
(Sluijs et al., 2006; Sluijs et al., 2009) and interpret the results following the currently
available TEX$_{86}$ proxy constraints.
Although contributions of isoGDGTs from land complicate TEX$_{86}$ paleothermometry
in some stratigraphic intervals, temperature was the dominant variable controlling
TEX$_{86}$ values. Background early Eocene SSTs exceed ~20 °C and peak warmth
occurred during the PETM and ETM2. However, uncertainty estimates of these SSTs
based on the non-analogue modern ocean, remains complex. Temperature constraints
from terrestrial vegetation support remarkable warmth in the study section and
elsewhere in the Arctic basin, notably coldest month mean temperatures around 10 °C
at least within the PETM and ETM2. If TEX$_{86}$-derived SSTs of ~20 °C truly represent
mean annual SSTs, the seasonal range of Arctic SST might have been in the order of
20 °C. If SST estimates are entirely skewed towards the summer season, seasonal
ranges in the order of 10 °C may be considered comparable to those simulated in future
ice-free Arctic Ocean scenarios.
We find abundant brGMGTs, which appear predominantly produced in the marine
realm at the study site. Their abundance increases during the PETM, likely due to sea



level rise and perhaps due to warming and a drop in seawater oxygen concentrations.
Although speculative, an increase in brGMGT methylation during the PETM may be a
function of temperature, but a relation between brGMGT distribution and
environmental parameters including temperature is yet to be confirmed.

**6. Data Availability**
All data is provided in the Supplement Table and will be included in the PANGAEA
database upon publication of this paper.

**7. Sample Availability**
Requests for materials can be addressed to A.Sluijs@uu.nl

**8. Author Contributions**
AS initiated the study, KGJN generated the data, JF modeled terrestrial contributions
of isoGDGTs based on published information and the new Crenarchaeol data of the
modern peat dataset, which was contributed by GNI. All authors contributed to the
interpretation of the data and AS wrote the paper with input from all authors.

**9. Competing Interests**
The authors declare no competing interests

**10. Acknowledgments**
We thank the ACEX scientific party for collaborations over the past 16 years, the
International Ocean Discovery Program (IODP) for access to ACEX samples and data,





and the Dutch Research Council (NWO) for their continued support to IODP. We thank
Linda van Roij for analytical support.
This research was funded by European Research Council Consolidator Grant 771497
awarded to AS and the Netherlands Earth System Science Centre, funded through a
Gravitation Grant by the Netherlands Ministry of Education, Culture and Science and
NWO. GNI acknowledges a GCRF Royal Society Dorothy Hodgkin Fellowship.

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



**Figure 1.** Location of ACEX Hole 4A within a paleogeographic reconstruction of the
Arctic region at the time of the PETM. Reconstruction made using gplates, using the
tectonic reconstruction of Seton et al. (2012, red shape is Lomonosov Ridge in this
reconstruction) using the paleomagnetic reference frame of Torsvik et al., (2012), and
modern coastlines.

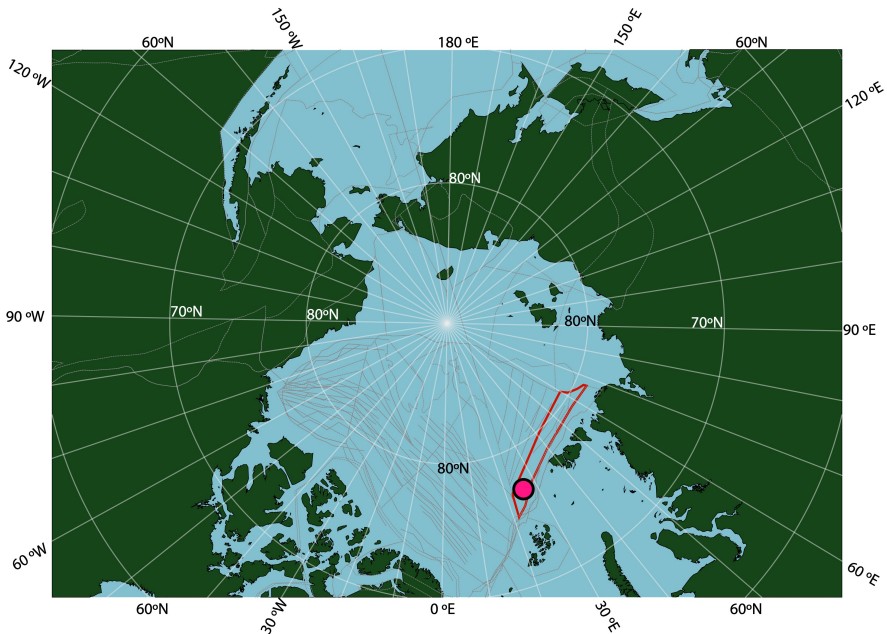







**Figure 2.** Chemical structures of the relevant isoGDGTs, brGDGTs and brGMGTs and
their terminology as described in this study. For the terminology of the brGMGTs, for
which the exact chemical structure is still unclear, we follow Baxter et al. (2019), since
we identify the same isomers (see Figure S2 for a chromatogram).

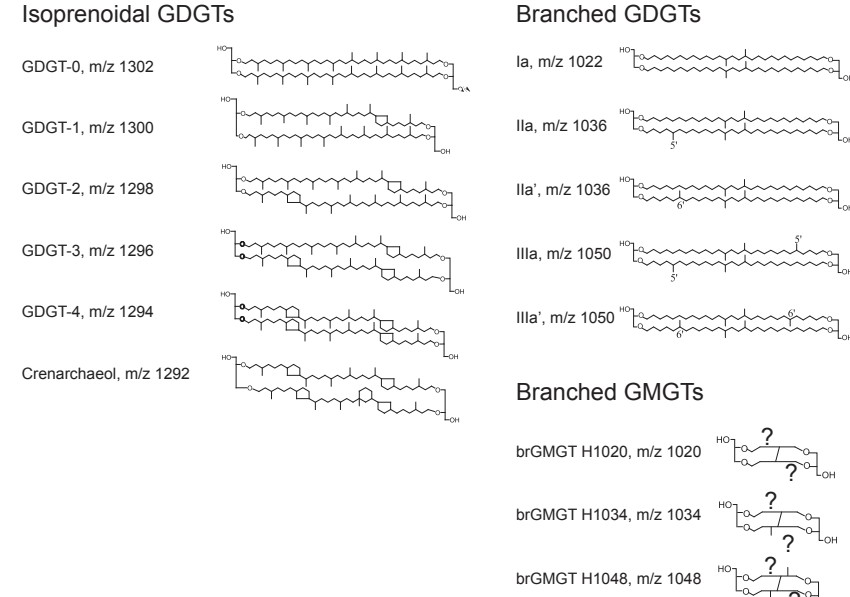






**Figure 3.** Comparison of the original GDGT dataset of the upper Paleocene and lower
Eocene of ACEX Hole 4A (Sluijs et al., 2006; Sluijs et al., 2009) and the new data
generated according to the latest chromatography protocols.

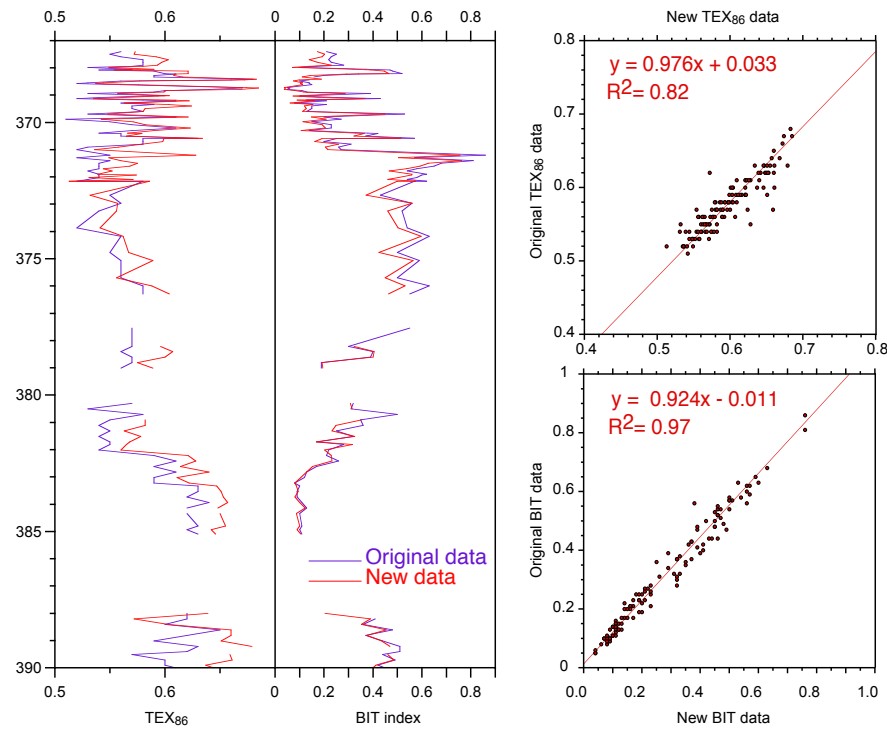




**Figure 4.** Comparison between BIT index values and $TEX_{86}$ for various intervals
spanning the upper Paleocene and lower Eocene of ACEX Hole 4A.

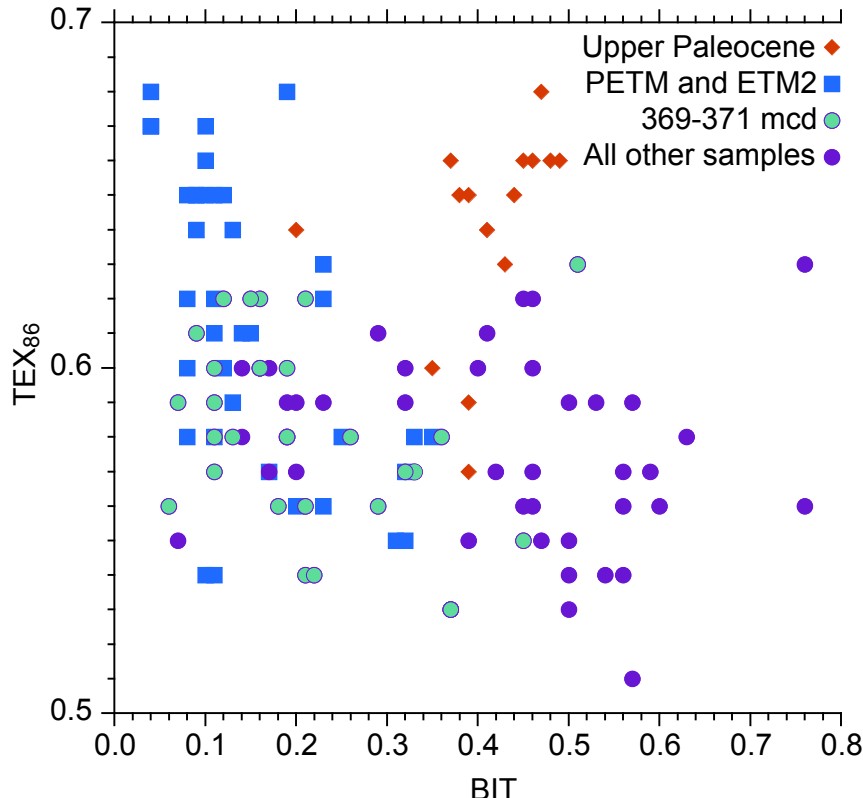




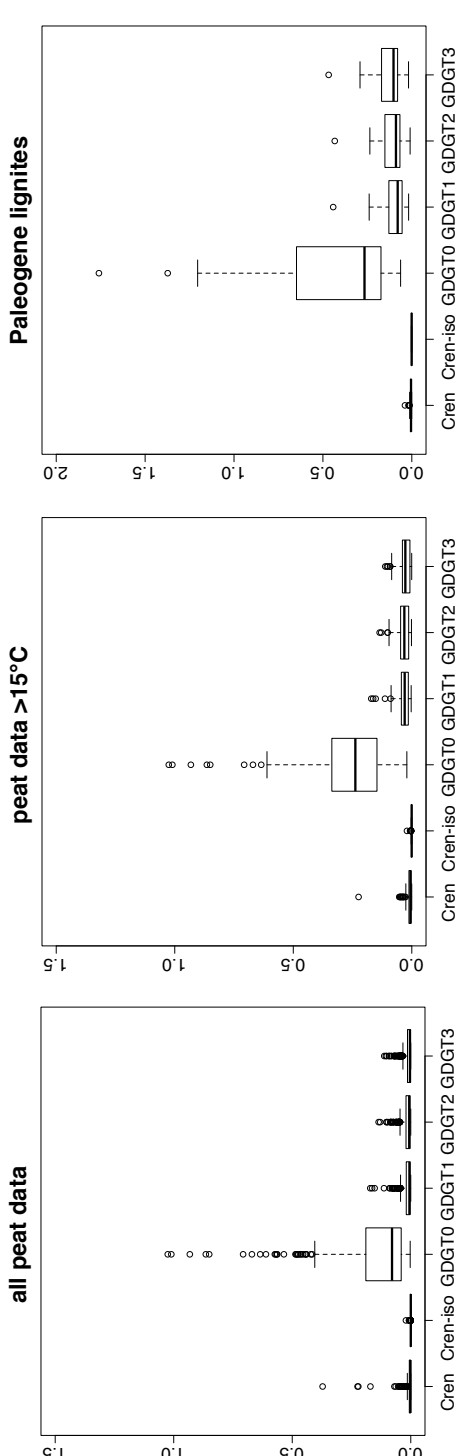

**Figure 5.** The distribution of various isoGDGTs relative to the total brGDGTs in modern peats and Paleogene lignites (Equation 9), used to assess potential isoGDGT contributions to the ACEX samples.


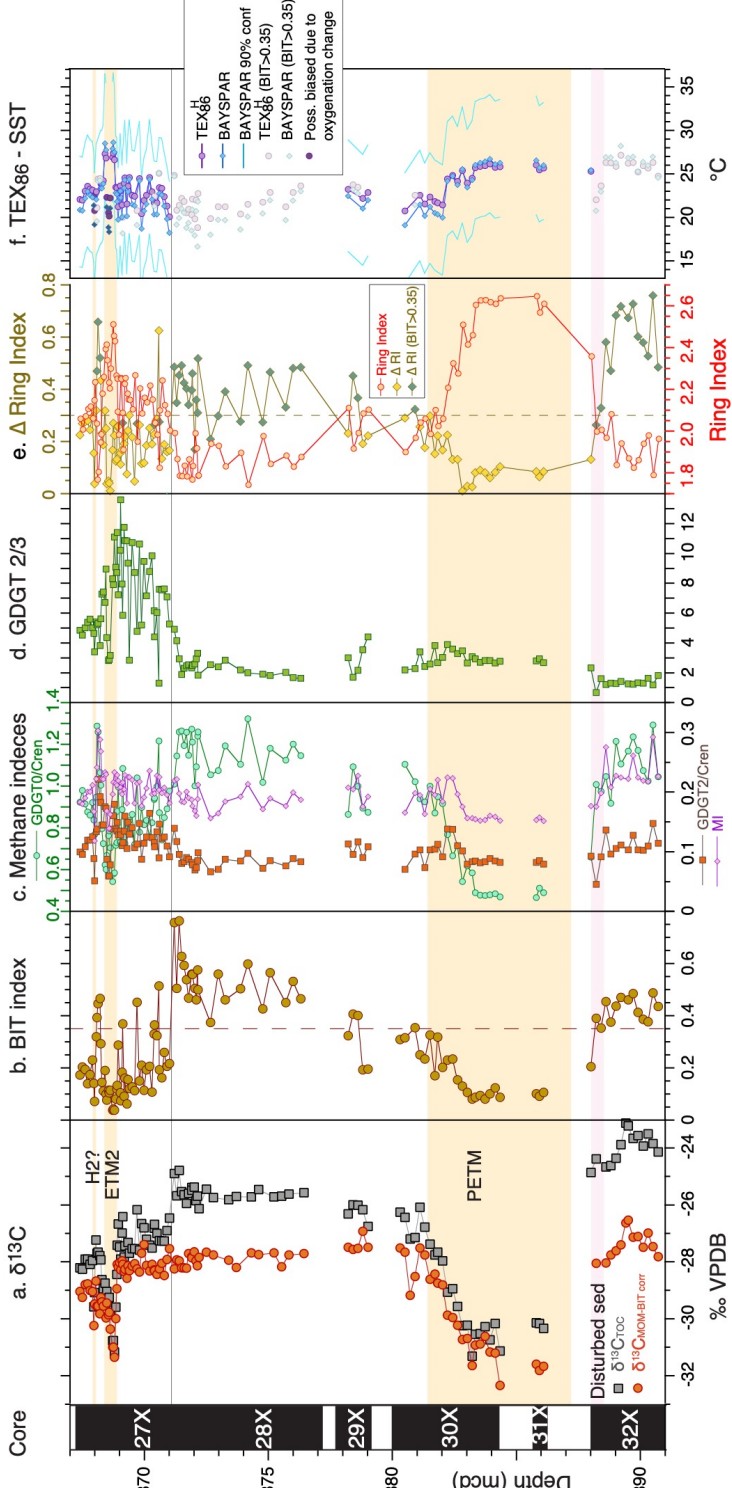

**Figure 6.** Branched and Isoprenoid GDGT records across the upper Paleocene and lower Eocene of ACEX Hole 4A. a. carbon isotope stratigraphy (total organic carbon record from Sluijs et al., 2006 and 2009; marine organic matter record from Sluijs and Dickens (2012)), b. BIT index (equation 2), c. indices indicative of anaerobic archaeal methanotrophy (MI index (equation 3) and GDGT-2/Crenarchaeol), and methanogenesis (GDGT-0/Crenarchaeol), d. GDGT2-GDGT3 ratio, e. Ring index (equation 5) and $\Delta$ Ring Index, f. $TEX_{86}$ (equation 1) calibrated to sea surface temperature using a non-linear calibration $TEX_{86}^H$ calibration (Kim et al., 2010) and the BAYSPAR method, which is based on a linear calibration (Tierney and Tingley, 2014).




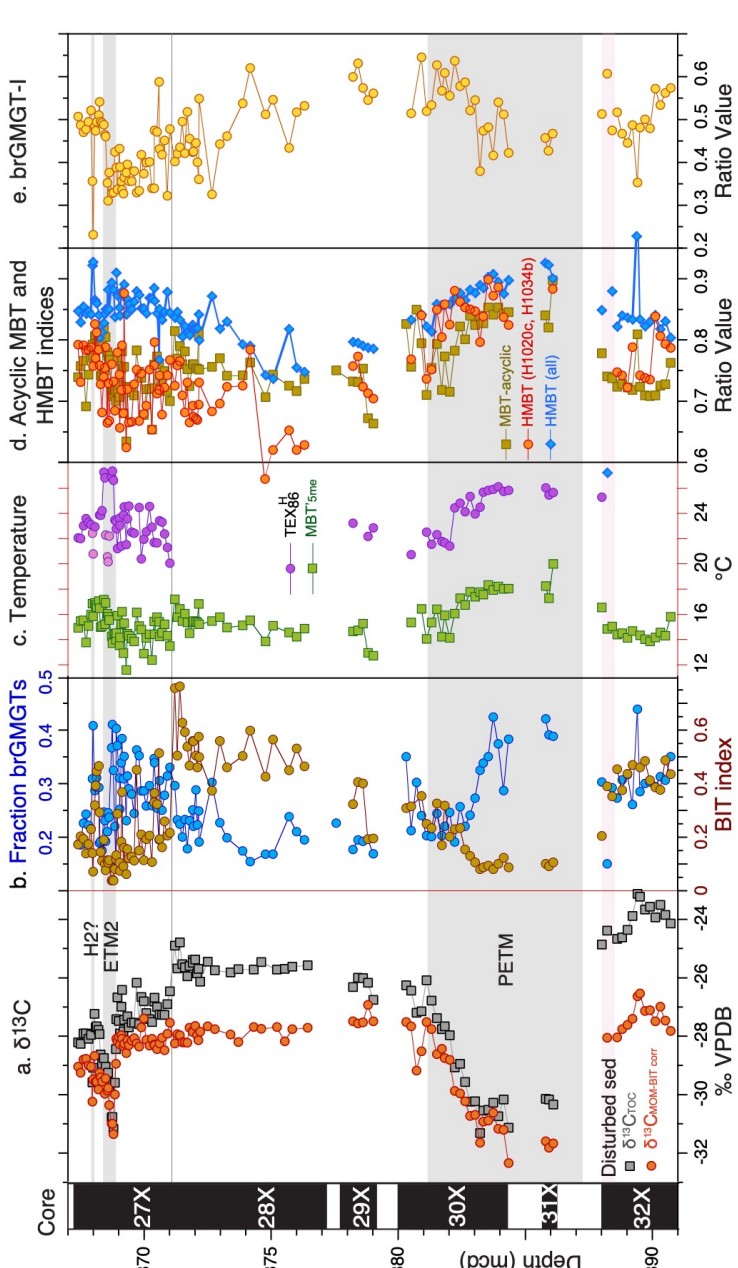

**Figure 7.** Branched GMGT records across the upper Paleocene and lower Eocene of ACEX Hole 4A. a. carbon isotope stratigraphy (total organic carbon record from Sluijs et al., 2006 and 2009; marine organic matter record from Sluijs and Dickens (2012)), b. fraction of brGMGTs of the total branched GDGTs and GMGTs and BIT index (equation 2), c. MBT$^H_{5me}$ record (Willard et al., 2019) and $TEX^H_{86}$, d. MBT*acyclic* (equation 6) and H-MBT based on all isomers detected with m/z 1020 and m/z 1034 (H-MBT all; equation 7) and based on H1020a and H1034b (H-MBT H1020a, H1034b, H1034c), e. brGMGT-1 record (equation 8).