# Peer review of "Late Paleocene – early Eocene Arctic Ocean Sea Surface Temperatures"

_Climate of the Past, 2020_

## Referee Comment (RC1) · Anonymous Referee #1 · 28 Apr 2020

In this work, Sluijs et al. revisit the ACEX early Paleogene sedimentary record to better examine the past temperature estimates from their earlier studies (Sluijs et al., 2006; 2009). I find this study important for paleoclimate, especially those who use GDGT-based proxy (e.g. TEX86), since there have been numerous improvements in constraining the proxy ('screening methods' such as Methane Index, Rind Index, GDGT-2/GDGT-3 etc) during the last decade. This implies that some of the earlier reported temperature reconstruction might contain data that are biased or unreliable. In addition, the Arctic region is an interesting study area which was very different from today. Many paleorecords and proxy studies indicate the Arctic was an ice-free warm environment, exceeding > 20 oC in SST. However, many climate models face challenge

in explaining the remarkably high temperature in the high-latitude region, unless the atmospheric pCO2 level was extremely high (e.g. 16 times higher than pre-industrial level; Pagani et al., 2014). This connects to the concept of meridional temperature gradient which has an important consequence in heat transport, ocean-atmospheric interaction, and global climate in the past.

Although there are several scientific interpretations to be addressed in the manuscript, I will save those for the next version, assuming some of them will be fixed in the revision. But I would like to comment on the data analysis here, which I think it should be addressed first.

Sluijs et al. used the previously analyzed samples which were stored for over a decade. As I am interested in GDGTs, I was curious how the old and new GDGT data would differ, although I assume the offset would be small if stored properly and measured in good condition of the HPLC/MS. Figure 3 shows the result and regression analysis between the old and measured GDGTs results. Both TEX86 and BIT look comparable. However, I found that there are few outliers in the TEX86 dataset from the supplementary data. I plotted all their new vs old TEX86, and the Rˆ2 value is lower to 0.66. Still comparable statistically, however, the authors did not mention about the outliers.

I appreciate the authors for providing their valuable dataset and kindly included the spreadsheet calculation for the readers to follow. For RI (ring index), however, I found that the calculations were all missing while it can be calculated from the dataset. I calculated again from their data but the values were slightly different. The maximum difference between the reported value (column BX) and the calculation I did is up to 0.11 RI unit. Although the difference is small, this would impact on some of the samples that have $\Delta$RI near 0.3, screening whether the data is reliable or unreliable near its cut-off value.

Overall, I suggest a moderate revision of the manuscript, especially in the data analysis first, before it can be accepted by CP. Also, the manuscript contained plagiarism (line

160-163) and many run-on sentences which made it difficult in absorbing the information when reading, therefore, I suggest a more improvement in the scientific writing for the next version.

Some specific comments are below:

Line 20-21: add "ACEX"

Line 20-52: the abstract seems to be too long and includes too much information of the study results in detail. Also, line 46-50 is just copied and pasted here from the main text (line 806-810).

Line 37: the background SSTs in early Eocene generally exceed

Line 71-77: run-on sentence: divided into two sentences

Line 77-84: I understand citing all the references to supplement, however, 17 citations are too overloaded in one sentence for the reader. I suggest organizing the citation to where they would belong. For example, link and cite Pagani et al. (2006) with "molecular fossils" which examined the $\delta$D of n-alkane addressing the hydrologic cycle. Or breakdown the sentence and cite only the important references.

Line 160-163: this sentence should be rephrased. It is exactly the same as written in Hollis et al. (2019) describing the BAYSPAR, but one word added here (plagiarism).

Line 169: add the TEX86 value range of which converted SSTs differs between linear & non-linear

Line 190-191: I suggest to remove "based on high BIT index value" and add the range of BIT results from the study after the equation.

Line 207: specify the GDGT. If just GDGT, does it mean both iso- and brGDGts?

Line 219-224: add the depth range of the deep contribution (Talyor et al., 2013) and also the reconstructed water depth of Site M00004A, meaning shallower shelf environment, to connect the interpretation of negligible deep source.

Line 233: use "[Crenarchaeol isomer]" for consistent compound name in all equations. this also implies for the names throughout the paper.

Line 234: "significant presence (or contribution) of anaerobic methanotrophy"

Line 242-243: provide references

Line 251: "Crenarchaeol isomer" for consistency of the compound name throughout the paper.

Line 299: I would rather suggest starting with 'brGMGTs' and supplement that this was previously reported as H-shaped brGDGTs, since the former is the major compound referred throughout the manuscript. Also, I suggest removing any description of 'H-shape brGDGTs' afterwards, as it makes it more confusing.

Line 344: the precision of TEX86 unit or converted SSTs unit?

Line 351-353: same comment with line 344. In addition, I am confused with what "both" labs means.

Line 409: interval should be between 371.0 to "369.0" mcd, based on Figure 4 and Sluijs et al. (2009),

Line 416-417: add the linear regression line in Figure 4 and supplement what "explaining 26 % of the variation" means

Line 428: I suggest to cite "Figure 6" in the first sentence, so the reader can easily compare the visualized data with the text, starting from the beginning of section or paragraph.

Line 442-452: Rather than directly moving on to the discussion of the method and result, I suggest to add a brief explanation of what lignite is and why lignite was used as the representative of terrestrial source for the readers to easily understand the concept.

Line 445: supplement how the absolute concentration is calculated (e.g. what standard used).

Line 467: "GDGT-2 and -3". Suggest describing the compounds be consistent throughout the paper.

Line 473-478: This is true based on the isoGDGT distributions of Paleogene lignite. The reported lignite samples' paleolatitudes are located within 57 °S to 48 °N, outside the Arctic region. Is there any lignite record from the Arctic that could be a more direct source to constrain the isoGDGTs distribution? If not, then how can this anomalous abundance of terrestrial isoGDGTs be explained in the Arctic where terrestrial input (especially from peats) is highly suggested while it has not been recorded elsewhere?

Line 486-487: add the threshold value of GDGT-2/Cren (Weijers et al., 2011), as it is shown as MI's cutoff in the following.

Line 492: I suggest the authors add a short interpretation of why these biomarker results are contrasted to the suitable depositional environment for abundant anaerobic archaea (methanotrophy and methanogen) which they indicated in the beginning of the section.

Line 508-510: interpreting BIT index with a distal position from the shoreline is problematic. Even in coastal marine or lacustrine settings, the BIT shows a large variation (Hopmans et al., 2004). Is the change of position interpreted from sea-level rise, similar to Sluijs et al. (2006)? Then what caused the sea-level rise (thermal expansion?) while the temperature proxy does not indicate significant warming?

Line 591: suggest the citation as "Figure 7b". This applies to other figure citations in the text to be more specific, when available, rather than just citing the whole figure. Another example is - line 606 to change to "Figure 7d"

Line 633-635: suggest to divide the two methods with (1) and (2), which the dashed line makes it confusing, and remove the linear/non-linear calibration description since

these are already explained previously.

Line 739: I find "lower temperature mean annual air temperature" very unclear.

Figure 1: (1) the word 'using' is used repeatedly – remove or organize with a different word (2) add gplate webpage link for the readers and reference (3) describe or indicate what the brownish lines in the map

Figure 2: (1) I suggest removing GDGT-4 since it is not discussed in the text nor measured in this study (see supplementary spreadsheet). Moreover, GDGT-4 is generally not included when calculating the relative fraction of isoGDGTs among the whole isoGDGTs pool. (2) add Crenarchaeol regioisomer's structure or note together with the Crenarchaeol (3) suggest changing "chemical structure" to "molecular structure"

Figure 5: (1) describe the "modern peats" into two in the caption. (2) describe what the box and line, error bar, circles indicate (3) add the number of samples for statistical meaning

Figure 7: (1) I suggest 7d and 7e switch the order, since it is the

Supplementary material Data table: (1) a lot of blanks in the sample data, as well as an unknown words or sample core names below the data seat (see row 153-157). (2) in "iGDGTs in peats" sheet, cite the references (3) in "Lignite crenarchaeol", Sluijs et al. reported the GDGTs (iso- and br-) data originally from Naaf et al. (2018) and their newly measured 'Cren. Isomer'. Here, I suggest the authors to report the other iso- and br-GDGTs abundances (here which I assume is HPLC/MS integrated peak area) together since they clearly mentioned in 'Material and Methods' that they re-analyzed the polar fraction of the lignite samples. Although I expect that this will not significantly change the result, still comparing only the newly measured 'Cren. Isomer' with reported GDGT dataset is not acceptable. This is because even measuring the same sample in the same method, the peak area can be different among interlaboratory measurements, the analytical parameter of the analytical instrument etc. In addition,

**CPD**

I suggest to add the calculations and results of the 'fraction of isoGDGTs' in all lignite samples. Lastly, minor comment on style of the table (e.g. missing cell borders, missing compound names) to be consistent. Describe 'n.d.' and 'b.d' too.
* * *

---

## Author Comment (AC1) · 1 May 2020

Dear editor,

We thank the Reviewer #1 for his/her comments. It seems to us that none of them present substantial criticism to any of our interpretations. Therefore, we will be able to swiftly incorporate all of his/her suggestions in our revised version, as we outline in our attached replies to all of his/her individual comments.

Sincerely, on behalf of all authors,

Appy Sluijs

[Figure]

Dear editor,

We thank the Reviewer #1 for his/her comments. It seems to us that none of them present substantial criticism to any of our interpretations. Therefore, we will be able to swiftly incorporate all of his/her suggestions in our revised version, as we outline below in our replies to all of his/her individual comments.

Sincerely, on behalf of all authors,

Appy Sluijs

Sluijs et al. used the previously analyzed samples which were stored for over a decade. As I am interested in GDGTs, I was curious how the old and new GDGT data would differ, although I assume the offset would be small if stored properly and measured in good condition of the HPLC/MS. Figure 3 shows the result and regression analysis between the old and measured GDGTs results. Both TEX86 and BIT look comparable. However, I found that there are few outliers in the TEX86 dataset from the supplementary data. I plotted all their new vs old TEX86, and the R´2 value is lower to 0.66. Still comparable statistically, however, the authors did not mention about the outliers.

*REPLY: These outliers represent data points for which the intensity of some isomers was insufficient in our reruns for proper quantification. For these 5 samples, TEX86 values were anomalously low as a consequence. These were the open fields in the spread sheet of the raw data but for clarity we have now marked them 'below detection' for the revised version of the manuscript. This further clarifies based on which data the 0.82 R^2 of Figure 3 is based.*

I appreciate the authors for providing their valuable dataset and kindly included the spreadsheet calculation for the readers to follow. For RI (ring index), however, I found that the calculations were all missing while it can be calculated from the dataset. I calculated again from their data but the values were slightly different. The maximum difference between the reported value (column BX) and the calculation I did is up to 0.11 RI unit. Although the difference is small, this would impact on some of the samples that have _RI near 0.3, screening whether the data is reliable or unreliable near its cutoff value.

*REPLY: We thank the reviewer very much for noticing this. The discrepancy was caused by an error in our excel calculations so that Cren isomer was not properly included. The numbers will be corrected in the revised supplement. The difference is indeed minor as the reviewer indicates and in fact it results in lower ΔRI and so we found no extra unreliable data points in our rescreening of the data.*

Overall, I suggest a moderate revision of the manuscript, especially in the data analysis first, before it can be accepted by CP. Also, the manuscript contained plagiarism (line 160-163) and many run-on sentences which made it difficult in absorbing the information when reading, therefore, I suggest a more improvement in the scientific writing for the next version.

*REPLY: We will make sure to reword this section and shorten sentences where necessary.*

Some specific comments are below:
Line 20-21: add "ACEX"
*REPLY: this shall be done*

Line 20-52: the abstract seems to be too long and includes too much information of the study results in detail. Also, line 46-50 is just copied and pasted here from the main text (line 806-810).
*REPLY: we will shorten the abstract significantly and avoid textual overlap with the rest of the text.*

Line 37: the background SSTs in early Eocene generally exceed
*REPLY: this shall be done*

**Fig. 1.**

---

## Referee Comment (RC2) · Tom Dunkley Jones (Referee) · 11 May 2020

This is an excellent and thorough reassessment of organic biomarker temperature records for the latest Paleocene and early Eocene, derived from sediments recovered from the central Arctic Ocean. As demonstrated within the manuscript, this time of peak Cenozoic warmth is a key interval of interest to the paleoclimate community. Considerable proxy data and climate model efforts are focusing on this interval to address questions of climate sensitivity and the persistent problem of extreme polar warmth, which is indicated by the proxy data but is still problematic for climate model simulations. The late Paleocene to early Eocene also includes multiple hyperthermal events

with millennial-scale onsets, which allow for the study of climate warming and ecosystem responses that approach the rates of modern environmental change. Two of these hyperthermal events are recovered within the ACEX record (PETM and ETM2).

The biomarker-based temperature data from the Lomonosov Ridge is a critical latitudinal "end-member" for an assessment of polar warmth during the latest Paleocene and early Eocene. The unusual GDGT assemblages extracted from these samples, and the initial efforts to use these to estimate sea surface temperatures – which by necessity were non-standard – left some concern within the community about their reliability as predictors of absolute temperatures. This study re-evaluates this critical record with new analyses, including of glycerol monoalkyl glycerol tetraethers (GMGTs), and places this new data within the context of the past decade of studies on the calibration and use of GDGT-based thermometry.

This study should be accepted for publication in Climates of the Past, although I do have one recommendation that I would like the authors to consider engaging with. Within this study they do a very thorough job of testing the potential controls and biases on GDGT assemblages using a range of indices and co-occurring markers for terrestrial-derived brGDGTs. The general aim of this is to screen GDGT assemblages, such that they can be separated into those that are formed in broadly analogous conditions to the modern marine system – and hence where the modern temperature-dependency of assemblage composition can be well-modelled by the modern core-top calibration - and those samples where the GDGT assemblage is significantly altered, by terrestrial input, methanogenesis or other processes, such that resultant estimates of SSTs may be biased. In their comprehensive treatment of this question of non-analogue behaviour and biases, my only recommendation is that the authors also consider the methods proposed by Eley et al. (Climates of the Past Discussions, 2019) for the detection of ancient GDGT assemblages that are significantly non-analogue to the modern calibration dataset. Below I include calculations of their Dnearest metric and OPTiMAL SST estimation for the new GDGT data presented by Sluijs et al. These

results confirm some of the key findings of Sluijs et al. – that the pre-PETM GDGT assemblages are anomalous relative to the modern calibration dataset (Dnearest> 1); that there two clear shifts towards GDGT assemblages more "typical" of the modern at ∼385.0m, and then again at ∼375.0m. There is also an interval after the PETM, where TEX86 based temperatures remain high (>20°C), whilst OPTiMAL temperatures are considerably lower, with values in the high single figures (∼375 to 371 mcd). Sluijs et al. show that pre-ETM2 GDGTs have high BIT indices (∼377 to ∼371 m) and do not consider TEX86 derived temperatures from this interval to be robust because of the potential bias from terrestrial-derived material. The OPTiMAL methodology, however, indicates that these pre-ETM2 GDGT assemblages are relatively closely analogous to GDGT assemblages in the modern core top data (Dnearest <0.5), and that these "near neighbours" are formed in locations with modern MAT SSTs below 10°C.

The Eley et al. (2019) methodology – and the one applied by me below (Figure 1) – includes all modern core top data within the Tierney & Tingley (2015) database, including Arctic data associated with SSTs <3°C. These data were excluded from the standard BAYSPAR calibration ("NoNorth" / "TT13" model of Tierney & Tingley, 2014), because in the Arctic region "TEX86 has a near-zero sensitivity to SST and therefore little predictability" and "incorporation of these data can negatively affect TEX86 predictability in the North Atlantic" (Tierney & Tingley, 2014). Although it would need to be tested – with OPTiMAL being run with and without these modern high-latitude data points and then applied to the ACEX core – it is possible that modern Arctic GDGT assemblages are the "nearest neighbours" of the pre-ETM2 GDGT assemblages, whereas above ∼371 mcd, assemblages shift to a more normal open marine assemblage, as inferred by Sluijs et al. on the basis of BIT indices. This may, in part, account for the significant warming suggested by OPTiMAL across this transition, and further work would be needed to investigate the inclusion or exclusion of modern Arctic GDGT assemblages in the modern calibration for OPTiMAL, and the ability to extract temperature information from these GDGT assemblages using proxy formulations other than the TEX86 index. Regardless of this, the consideration of the OPTiMAL approach confirms - through an independent approach that is agnostic about the form or "model choice", of the GDGT – SST relationship - that Arctic SSTs around ETM2 were in the region of ∼20°C (OPTiMAL) or higher (TEX86H, BAYSPAR).

Section 2.1 and especially Lines 145 – 147: suggest that culture and mesocosm experiments and surface sediment data indicate a linear relationship but without a clear citation of these studies. Rather, the citations seem to be of the studies that demonstrate a deviation from linearity. As the authors implicitly acknowledge - with statements like "suggest a linear relation" (line 146) or "assumes a linear relationship" (line 160) - the most appropriate form of the TEX86 – SST relationship is uncertain, with current calibration models making some degree of assumption about the best fit relationship between core top TEX86 data and SSTs. I would suggest a slight rephrase to acknowledge this uncertainty and appropriate citations to back up any arguments made about the form of the relationship. There is extensive discussion of the assumptions that can be made about the form of the TEX86 – SST relationship within the online discussion to Eley et al. (2019) that address this issue, between those who argue for an assumed linear response (Tierney) and those who question this assumption (Eley et al.) – some of the relevant response to Tierney quoted below from Eley et al. (https://www.clim-past-discuss.net/cp-2019-60/cp-2019-60-AC1-supplement.pdf):

"We agree that there is a basic underlying trend for more rings within GDGT structures at higher temperatures (Zhang et al. 2015; Qin et al., 2015). What we dispute is that this translates into a simple linear model at the community scale (core top calibration dataset), or is yet reproduced with consistency between strains in laboratory cultures, including the temperature-dependence of GDGT ring numbers within the marine, mesophilic Thaumarchaeota in Marine Group 1 (broadly equivalent to the old Crenarchaeota Group 1) (Eilling et al., 2015; Qin et al., 2015; Wuchter et al., 2007). Wuchter et al. (2004) and Schouten et al. (2007) show a compiled linear calibration of TEX86 against incubation temperature (up to 40°C in the case of Schouten et al., 2007) based on strains that were enriched from surface seawater collected from the

North Sea and Indian Ocean respectively. Like Qin et al., (2015) we note the non-linear nature of the individual experiments in Wuchter et al., 2004 (see Wuchter et al., 2004 Fig. 5). Moreover, the relatively lower Cren' in these studies yield a very different intercept and slope (compared to core-top calibrations e.g. Kim et al. 2010) meaning that the resulting calibrations for TEX86 cannot be applied to core-tops. This was recognised by Kim et al. (2010), who state "but we may speculate that Marine Group I Crenarchaeota species in the enrichment cultures are not completely representative of those occurring in nature. . .

. . .As we state above, although we agree that there is a basic underlying trend of increasing ring number with increasing growth temperature, we do not agree that this is well enough known to be quantified into a "basic relationship" that can be "enforced" as a particular model form. Rather, there is uncertainty in the appropriate form of the relationship even within the modern calibration data (see Kim et al. 2010) which becomes substantial beyond the calibration range. The spatial structuring of residuals in global models of modern TEX86 temperature dependence (Tierney & Tingley, 2014) and clear structuring of residuals with temperature in our and other GDGT- temperature calibrations, are likely indications of transitions in the ecology, community make-up or habitat of modern GDGT producers that are not well constrained. We argue that this complexity in the GDGT temperature responses in the modern oceans should be grounds for caution when applying empirical models from the modern to ancient conditions, especially when working with the subset of ancient assemblage data for which there is no modern analogue."

Eley, Y. L., Thompson, W., Greene, S. E., Mandel, I., Edgar, K., Bendle, J. A., and Dunkley Jones, T.: OPTiMAL: A new machine learning approach for GDGT-based palaeothermometry, Clim. Past Discuss., https://doi.org/10.5194/cp-2019-60, in review, 2019.

Tierney, J. E., and Tingley, M. P.: A Bayesian, spatially-varying calibration model for the TEX86 proxy. Geochimica et Cosmochimica Acta, 127, 83-106, 2014.

Tierney, J. E., and Tingley, M. P.: A TEX86 surface sediment database and extended Bayesian calibration, Scientific data, 2, 150029, 2015.

[Figure]

[Figure]

**Fig. 1.** Figure 1. Left: OPTiMAL-derived SSTs with one standard deviation error bars; data in grey fail the Dnearest test of Eley et al. (2019) (Dnearest > 0.5). Right: Dnearest values through the succession.

---

## Author Comment (AC2) · 29 May 2020

This is an excellent and thorough reassessment of organic biomarker temperature records for the latest Paleocene and early Eocene, derived from sediments recovered from the central Arctic Ocean. As demonstrated within the manuscript, this time of peak Cenozoic warmth is a key interval of interest to the paleoclimate community. Considerable proxy data and climate model efforts are focusing on this interval to address questions of climate sensitivity and the persistent problem of extreme polar warmth, which is indicated by the proxy data but is still problematic for climate model simulations. The late Paleocene to early Eocene also includes multiple hyperthermal events with millennial-scale onsets, which allow for the study of climate warming and ecosystem responses that approach the rates of modern environmental change. Two of these hyperthermal events are recovered within the ACEX record (PETM and ETM2). The biomarker-based temperature data from the Lomonosov Ridge is a critical latitudinal "end-member" for an assessment of polar warmth during the latest Paleocene and early Eocene. The unusual GDGT assemblages extracted from these samples, and the initial efforts to use these to estimate sea surface temperatures – which by necessity were non-standard – left some concern within the community about their reliability as predictors of absolute temperatures. This study re-evaluates this critical record with new analyses, including of glycerol monoalkyl glycerol tetraethers (GMGTs), and places this new data within the context of the past decade of studies on the calibration and use of GDGT-based thermometry.

Reply: We thank Dr. Dunkley-Jones for his support of our work.

**Point 1.** This study should be accepted for publication in Climates of the Past, although I do have one recommendation that I would like the authors to consider engaging with. Within this study they do a very thorough job of testing the potential controls and biases on GDGT assemblages using a range of indices and co-occurring markers for terrestrial-derived brGDGTs. The general aim of this is to screen GDGT assemblages, such that they can be separated into those that are formed in broadly analogous conditions to the modern marine system – and hence where the modern temperature dependency of assemblage composition can be well-modelled by the modern core-top calibration - and those samples where the GDGT assemblage is significantly altered, by terrestrial input, methanogenesis or other processes, such that resultant estimates of SSTs may be biased. In their comprehensive treatment of this question of non analogue behaviour and biases, my only recommendation is that the authors also consider the methods proposed by Eley et al. (Climates of the Past Discussions, 2019) for the detection of ancient GDGT assemblages that are significantly non-analogue to the modern calibration dataset. Below I include calculations of their Dnearest metric and OPTiMAL SST estimation for the new GDGT data presented by Sluijs et al. These results confirm some of the key findings of Sluijs et al. – that the pre-PETM GDGT assemblages are anomalous relative to the modern calibration dataset (Dnearest> 1); that there two clear shifts towards GDGT assemblages more "typical" of the modern at _385.0m, and then again at _375.0m. There is also an interval after the PETM, where TEX86 based temperatures remain high (>20_C), whilst OPTiMAL temperatures are considerably lower, with values in the high single figures (_375 to 371 mcd). Sluijs et al. show that pre-ETM2 GDGTs have high BIT indices (_377 to _371 m) and do not consider TEX86 derived temperatures from this interval to be robust because of the potential bias from terrestrial-derived material. The OPTiMAL methodology, however, indicates that these pre-ETM2 GDGT assemblages are relatively closely analogous to GDGT assemblages in the modern core top data (Dnearest <0.5), and that these "near neighbours" are formed in locations with modern MAT SSTs below 10_C.

The Eley et al. (2019) methodology – and the one applied by me below (Figure 1) – includes all modern core top data within the Tierney & Tingley (2015) database, including Arctic data associated with SSTs <3_C. These data were excluded from the standard BAYSPAR calibration ("NoNorth" /

"TT13" model of Tierney & Tingley, 2014), because in the Arctic region "TEX86 has a near-zero sensitivity to SST and therefore little predictability" and "incorporation of these data can negatively affect TEX86 predictability in the North Atlantic" (Tierney & Tingley, 2014). Although it would need to be tested – with OPTiMAL being run with and without these modern high-latitude data points and then applied to the ACEX core – it is possible that modern Arctic GDGT assemblages are the "nearest neighbours" of the pre-ETM2 GDGT assemblages, whereas above _371 mcd, assemblages shift to a more normal open marine assemblage, as inferred by Sluijs et al. on the basis of BIT indices. This may, in part, account for the significant warming suggested by OPTiMAL across this transition, and further work would be needed to investigate the inclusion or exclusion of modern Arctic GDGT assemblages in the modern calibration for OPTiMAL, and the ability to extract temperature information from these GDGT assemblages using proxy formulations other than the TEX86 index. Regardless of this, the consideration of the OPTiMAL approach con- firms - through an independent approach that is agnostic about the form or "model choice", of the GDGT – SST relationship - that Arctic SSTs around ETM2 were in the region of _20_C (OPTiMAL) or higher (TEX86H, BAYSPAR).

**Reply to point 1.** We are aware of the Eley et al. manuscript and their new indicators are certainly potentially interesting for this paper. As far as we can see on the CP website, however, this paper is under evaluation and, considering the online discussion, may be subject to significant revisions. This compromises the use of this method in its present form.

We have nevertheless considered the Dnearest and OPTiMAL records kindly provided by the reviewer. Certainly, as the reviewer indicates, some of the results are consistent with our results (e.g. during the latest Paleocene) and could in principle be used as support for our statements. Other aspects are inconsistent with what we find. For example, the interval 377-371 m discussed by the reviewer is particularly interesting. Although the Dnearest metric suggests that TEX$_{86}$ values are robust, the delta Ring Index, a simple metric to evaluate whether GDGT distributions are similar to modern ocean core top data (Zhang et al., 2011), clearly indicates the GDGT assemblage is compromised for reliable TEX86 paleothermometry. The BIT index (>0.4 to 0.8), the palynological assemblage (dominated by terrestrial organic matter) the dinoflagellate cyst assemblage (almost completely freshwater-tolerant) and bulk organic carbon stable carbon isotope ratios are also inconsistent with a dominant marine source of the organic matter (Sluijs et al., 2008; Sluijs et al., 2009; Sluijs and Dickens, 2012). This would call into question the reliability of the Dnearest metric as an indicator of "normal" marine isoGDGT assemblages, even if they are consistent with some of the modern core top data. Therefore, it seems to us that any discussion on this topic would rather serve as a test to the as yet unpublished new methods proposed by Eley et al. rather than contributing to the main goal of our paper, to reconstruct late Paleocene – early Eocene Arctic paleotemperature.

Collectively, we therefore prefer to not discuss these results at this point.

**Point 2.** Section 2.1 and especially Lines 145 – 147: suggest that culture and mesocosm experiments and surface sediment data indicate a linear relationship but without a clear citation of these studies. Rather, the citations seem to be of the studies that demonstrate a deviation from linearity. As the authors implicitly acknowledge - with statements like "suggest a linear relation" (line 146) or "assumes a linear relationship" (line 160) - the most appropriate form of the TEX86 – SST relationship is uncertain, with current calibration models making some degree of assumption about the best fit relationship between core top TEX86 data and SSTs. I would suggest a slight rephrase to acknowledge this uncertainty and appropriate citations to back up any arguments made about the form of the relationship. There is extensive discussion of the assumptions that can be made about the form of the TEX86 – SST relationship within the online discussion to Eley et al. (2019) that address this issue, between those who argue for an assumed linear response (Tierney) and those who question this

assumption (Eley et al.) – some of the relevant response to Tierney quoted below from Eley et al. (https://www.climpast-discuss.net/cp-2019-60/cp-2019-60-AC1-supplement.pdf):

"We agree that there is a basic underlying trend for more rings within GDGT structures at higher temperatures (Zhang et al. 2015; Qin et al., 2015). What we dispute is that this translates into a simple linear model at the community scale (core top calibration dataset), or is yet reproduced with consistency between strains in laboratory cultures, including the temperature-dependence of GDGT ring numbers within the marine, mesophilic Thaumarchaeota in Marine Group 1 (broadly equivalent to the old Crenarchaeota Group 1) (Eilling et al., 2015; Qin et al., 2015; Wuchter et al., 2007). Wuchter et al. (2004) and Schouten et al. (2007) show a compiled linear calibration of TEX86 against incubation temperature (up to 40_C in the case of Schouten et al., 2007) based on strains that were enriched from surface seawater collected from the North Sea and Indian Ocean respectively. Like Qin et al., (2015) we note the nonlinear nature of the individual experiments in Wuchter et al., 2004 (see Wuchter et al., 2004 Fig. 5). Moreover, the relatively lower Cren' in these studies yield a very different intercept and slope (compared to core-top calibrations e.g. Kim et al. 2010) meaning that the resulting calibrations for TEX86 cannot be applied to core-tops. This was recognised by Kim et al. (2010), who state "but we may speculate that Marine Group I Crenarchaeota species in the enrichment cultures are not completely representative of those occurring in nature…

…As we state above, although we agree that there is a basic underlying trend of increasing ring number with increasing growth temperature, we do not agree that this is well enough known to be quantified into a "basic relationship" that can be "enforced" as a particular model form. Rather, there is uncertainty in the appropriate form of the relationship even within the modern calibration data (see Kim et al. 2010) which becomes substantial beyond the calibration range. The spatial structuring of residuals in global models of modern TEX86 temperature dependence (Tierney & Tingley, 2014) and clear structuring of residuals with temperature in our and other GDGT- temperature calibrations, are likely indications of transitions in the ecology, community make-up or habitat of modern GDGT producers that are not well constrained. We argue that this complexity in the GDGT temperature responses in the modern oceans should be grounds for caution when applying empirical models from the modern to ancient conditions, especially when working with the subset of ancient assemblage data for which there is no modern analogue."

**Reply to point 2.** We fully agree with the reviewer here. In fact, some of us have argued for the possibility of a non-linear relation ourselves in a recent paper (Cramwinckel et al., 2018). We will adapt the text in section 2.1 to fully reflect the current status of the discussion (e.g., Hollis et al., 2019).

**References**

Cramwinckel, M. J., Huber, M., Kocken, I. J., Agnini, C., Bijl, P. K., Bohaty, S. M., Frieling, J., Goldner, A., Hilgen, F. J., Kip, E. L., Peterse, F., van der Ploeg, R., Röhl, U., Schouten, S., and Sluijs, A.: Synchronous tropical and polar temperature evolution in the Eocene, Nature, 559, 382-386, 10.1038/s41586-018-0272-2, 2018.

Hollis, C. J., Dunkley Jones, T., Anagnostou, E., Bijl, P. K., Cramwinckel, M. J., Cui, Y., Dickens, G. R., Edgar, K. M., Eley, Y., Evans, D., Foster, G. L., Frieling, J., Inglis, G. N., Kennedy, E. M., Kozdon, R., Lauretano, V., Lear, C. H., Littler, K., Lourens, L., Meckler, A. N., Naafs, B. D. A., Pälike, H., Pancost, R. D., Pearson, P. N., Röhl, U., Royer, D. L., Salzmann, U., Schubert, B. A., Seebeck, H., Sluijs, A., Speijer, R. P., Stassen, P., Tierney, J., Tripati, A., Wade, B., Westerhold, T., Witkowski, C., Zachos, J. C., Zhang, Y. G., Huber, M., and Lunt, D. J.: The DeepMIP contribution to PMIP4: methodologies for selection, compilation and analysis of latest Paleocene and early Eocene climate proxy data, incorporating version 0.1 of the DeepMIP database, Geosci. Model Dev., 12, 3149-3206, 10.5194/gmd-12-3149-2019, 2019.

Sluijs, A., Röhl, U., Schouten, S., Brumsack, H.-J., Sangiorgi, F., Sinninghe Damsté, J. S., and Brinkhuis, H.: Arctic late Paleocene–early Eocene paleoenvironments with special emphasis on the Paleocene-Eocene thermal maximum (Lomonosov Ridge, Integrated Ocean Drilling Program Expedition 302), Paleoceanography, 23, PA1S11, doi:10.1029/2007PA001495, 2008.

Sluijs, A., Schouten, S., Donders, T. H., Schoon, P. L., Röhl, U., Reichart, G. J., Sangiorgi, F., Kim, J.-H., Sinninghe Damsté, J. S., and Brinkhuis, H.: Warm and Wet Conditions in the Arctic Region during Eocene Thermal Maximum 2, Nature Geoscience, 2, 777-780, 2009.

Sluijs, A., and Dickens, G. R.: Assessing offsets between the $\delta^{13}C$ of sedimentary components and the global exogenic carbon pool across Early Paleogene carbon cycle perturbations, Global Biogeochemical Cycles, 26, GB4005, doi:10.1029/2011GB004224, 2012.

---

## Author Comment (AC3) · 29 May 2020

Dear editor,

We thank the Reviewer #1 for his/her comments. It seems to us that none of them present substantial criticism to any of our interpretations. Therefore, we will be able to swiftly incorporate all of his/her suggestions in our revised version, as we outline below in our replies to all of his/her individual comments.

Sincerely, on behalf of all authors,

Appy Sluijs

Sluijs et al. used the previously analyzed samples which were stored for over a decade. As I am interested in GDGTs, I was curious how the old and new GDGT data would differ, although I assume the offset would be small if stored properly and measured in good condition of the HPLC/MS. Figure 3 shows the result and regression analysis between the old and measured GDGTs results. Both TEX86 and BIT look comparable. However, I found that there are few outliers in the TEX86 dataset from the supplementary data. I plotted all their new vs old TEX86, and the R^2 value is lower to 0.66. Still comparable statistically, however, the authors did not mention about the outliers.

*REPLY: These outliers represent data points for which the intensity of some isomers was insufficient in our reruns for proper quantification. For these 5 samples, TEX86 values were anomalously low as a consequence. These were the open fields in the spread sheet of the raw data but for clarity we have now marked them 'below detection' for the revised version of the manuscript. This further clarifies based on which data the 0.82 R^2 of Figure 3 is based.*

I appreciate the authors for providing their valuable dataset and kindly included the spreadsheet calculation for the readers to follow. For RI (ring index), however, I found that the calculations were all missing while it can be calculated from the dataset. I calculated again from their data but the values were slightly different. The maximum difference between the reported value (column BX) and the calculation I did is up to 0.11 RI unit. Although the difference is small, this would impact on some of the samples that have _RI near 0.3, screening whether the data is reliable or unreliable near its cutoff value.

*REPLY: We thank the reviewer very much for noticing this. The discrepancy was caused by an error in our excel calculations so that Cren isomer was not properly included. The numbers will be corrected in the revised supplement. The difference is indeed minor as the reviewer indicates and in fact it results in lower ΔRI and so we found no extra unreliable data points in our rescreening of the data.*

Overall, I suggest a moderate revision of the manuscript, especially in the data analysis first, before it can be accepted by CP. Also, the manuscript contained plagiarism (line 160-163) and many run-on sentences which made it difficult in absorbing the information when reading, therefore, I suggest a more improvement in the scientific writing for the next version.

*REPLY: We will make sure to reword this section and shorten sentences where necessary.*

Some specific comments are below:
Line 20-21: add "ACEX"
*REPLY: this shall be done*

Line 20-52: the abstract seems to be too long and includes too much information of the study results in detail. Also, line 46-50 is just copied and pasted here from the main text (line 806-810).
*REPLY: we will shorten the abstract significantly and avoid textual overlap with the rest of the text.*

Line 37: the background SSTs in early Eocene generally exceed
*REPLY: this shall be done*

Line 71-77: run-on sentence: divided into two sentences
*REPLY: this shall be done*

Line 77-84: I understand citing all the references to supplement, however, 17 citations are too overloaded in one sentence for the reader. I suggest organizing the citation to where they would belong. For example, link and cite Pagani et al. (2006) with "molecular fossils" which examined the _D of n-alkane addressing the hydrologic cycle. Or breakdown the sentence and cite only the important references.
*REPLY: we will cite three of the early papers that showed the potential for follow-up work.*

Line 160-163: this sentence should be rephrased. It is exactly the same as written in Hollis et al. (2019) describing the BAYSPAR, but one word added here (plagiarism).
*REPLY: this shall be reworded*

Line 169: add the TEX86 value range of which converted SSTs differs between linear & non-linear
*REPLY: This has been extensively discussed in the literature and it seems that the divergence occurs close to the maximum value in the calibration dataset (ca. 0.70), which we will include here.*

Line 190-191: I suggest to remove "based on high BIT index value" and add the range of BIT results from the study after the equation.
*REPLY: this shall be done and we will add a sentence reporting on a previous subjectively defined threshold assigned on the study section (Sluijs et al., 2009).*

Line 207: specify the GDGT. If just GDGT, does it mean both iso- and brGDGts?
*REPLY: this shall be specified to isoGDGT*

Line 219-224: add the depth range of the deep contribution (Talyor et al., 2013) and also the reconstructed water depth of Site M00004A, meaning shallower shelf environ- ment, to connect the interpretation of negligible deep source.
*REPLY: This will be done (>1000 m)*

Line 233: use "[Crenarchaeol isomer]" for consistent compound name in all equations. this also implies for the names throughout the paper.
*REPLY: This will be done*

Line 234: "significant presence (or contribution) of anaerobic methanotrophy"
*REPLY: This will be done*

Line 242-243: provide references
*REPLY: This will be done*

Line 251: "Crenarchaeol isomer" for consistency of the compound name throughout the paper.
*REPLY: We will stick to this wording throughout.*

Line 299: I would rather suggest starting with 'brGMGTs' and supplement that this was previously reported as H-shaped brGDGTs, since the former is the major compound referred throughout the manuscript. Also, I suggest removing any description of 'Hshape brGDGTs' afterwards, as it makes it more confusing.
*REPLY: This suggestion will be followed*

Line 344: the precision of TEX86 unit or converted SSTs unit?
*REPLY: We will explain that this regards the uncertainty calculated to the SST domain.*

Line 351-353: same comment with line 344. In addition, I am confused with what "both" labs means.
*REPLY: We will delete this confusing sentence.*

Line 409: interval should be between 371.0 to "369.0" mcd, based on Figure 4 and Sluijs et al. (2009),
*REPLY: Indeed, thank you for noticing.*

Line 416-417: add the linear regression line in Figure 4 and supplement what "explaining 26 % of the variation" means
*REPLY: This will be done and we will clarify the statement on variation; this number is taken directly from the R^2 (0.26) of the linear regression.*

Line 428: I suggest to cite "Figure 6" in the first sentence, so the reader can easily compare the visualized data with the text, starting from the beginning of section or paragraph.
*REPLY: This will be done*

Line 442-452: Rather than directly moving on to the discussion of the method and result, I suggest to add a brief explanation of what lignite is and why lignite was used as the representative of terrestrial source for the readers to easily understand the concept.
*REPLY: This will be done. We use the peat and lignite databases because they represent comprehensive datasets and therefore allow a rough calculation of the potential isoGDGT contribution.*

Line 445: supplement how the absolute concentration is calculated (e.g. what standard used).
*REPLY: The reference to absolute concentration was incorrect. It will be changed to raw signal intensity.*

Line 467: "GDGT-2 and -3". Suggest describing the compounds be consistent throughout the paper.
*REPLY: This will be done*

Line 473-478: This is true based on the isoGDGT distributions of Paleogene lignite. The reported lignite samples' paleolatitudes are located within 57 _S to 48 _N, outside the Arctic region. Is there any lignite record from the Arctic that could be a more direct source to constrain the isoGDGTs distribution? If not, then how can this anomalous abundance of terrestrial isoGDGTs be explained in the Arctic where terrestrial input (especially from peats) is highly suggested while it has not been recorded elsewhere?
*REPLY: We are not aware of any study that describes such high abundances of GDGT-3 nor a study that describes GDGT distributions from a northern high latitude Paleogene lignite, such as those described by Suan et al. (2017). In addition, we do not argue that peats are the main contributor to the terrestrial isoGDGT contribution. We merely include this exercise as a crude model for the potential terrestrial contribution to the isoGDGT pool in our ACEX samples, as we will better explain in the next version of the manuscript. Ideally, the analyses we perform here are also conducted using the abundance of isoGDGTs relative to brGDGTs in mineral soils to provide an even more complete picture, but those paired data are not available.*

Line 486-487: add the threshold value of GDGT-2/Cren (Weijers et al., 2011), as it is shown as MI's cutoff in the following.
*REPLY: As far as we are aware, a formal threshold or cut off was never defined, but our values are clearly within the safe range of values described by Weijers et al. (2011), which is what we indicated here.*

Line 492: I suggest the authors add a short interpretation of why these biomarker results are contrasted to the suitable depositional environment for abundant anaerobic archaea (methanotrophy and methanogen) which they indicated in the beginning of the section.
*REPLY: This will be done*

Line 508-510: interpreting BIT index with a distal position from the shoreline is problematic. Even in coastal marine or lacustrine settings, the BIT shows a large variation (Hopmans et al., 2004). Is the change of position interpreted from sea-level rise, similar to Sluijs et al. (2006)? Then what caused the sea-level rise (thermal expansion?) while the temperature proxy does not indicate significant warming?

*REPLY: We do not only rely on the BIT index but also on palynological evidence that is consistent with a relative drop in terrestrial organic matter contribution. Relative sea level rise is clearly the simplest explanation for the observed changes. Sluijs et al. (2006) described sea level rise during the PETM that was later shown to be eustatic (Sluijs et al., 2008a). The interval described here regards an episode of relative sea level rise some time before ETM2. We are not aware of literature that has seen similar relative sea level rise elsewhere so we presume this relative sea level rise was of local, perhaps tectonic, origin. We will rephrase as follows:*

*"At ~371.2 mcd a drop in BIT index and a change in the palynological assemblages corresponds to an interval of greenish sediment, suggestive of pronounced amounts of glauconite. These changes are consistent with local relative sea level rise, causing a somewhat more distal position relative to the shoreline. However, the sediment remains dominantly siliciclastic and organic terrestrial components, particularly pollen and spores, remain abundant still indicating a shallow setting (Sluijs et al., 2008a; Sluijs et al., 2008b)."*

Line 591: suggest the citation as "Figure 7b". This applies to other figure citations in the text to be more specific, when available, rather than just citing the whole figure. Another example is - line 606 to change to "Figure 7d"
*REPLY: This will be done*

Line 633-635: suggest to divide the two methods with (1) and (2), which the dashed line makes it confusing, and remove the linear/non-linear calibration description since these are already explained previously.
*REPLY: We will include the 1) and 2) suggestion but we choose to keep the reminder to the reader regarding the linear vs non-linear calibrations.*

Line 739: I find "lower temperature mean annual air temperature" very unclear.
*REPLY: We will delete the first 'temperature' to solve this issue.*

Figure 1: (1) the word 'using' is used repeatedly – remove or organize with a different word (2) add gplate webpage link for the readers and reference (3) describe or indicate what the brownish lines in the map
*REPLY: We will rephrase the caption accordingly*

Figure 2: (1) I suggest removing GDGT-4 since it is not discussed in the text nor measured in this study (see supplementary spreadsheet). Moreover, GDGT-4 is generally not included when calculating the relative fraction of isoGDGTs among the whole isoGDGTs pool. (2) add Crenarchaeol regioisomer's structure or note together with the Crenarchaeol (3) suggest changing "chemical structure" to "molecular structure"
*REPLY: We will change accordingly. Specifically, we will remove isoGDGT-4 from the figure as suggested. We note in the caption of figure 2 that the Crenarchaeol isomer differs from Crenarchaeol in the stereochemistry of a cyclopentane moiety (Sinninghe Damsté et al., 2018), and replace 'chemical' with 'molecular' as suggested.*

Figure 5: (1) describe the "modern peats" into two in the caption. (2) describe what the box and line, error bar, circles indicate (3) add the number of samples for statistical meaning
*REPLY: This will be done*

Figure 7: (1) I suggest 7d and 7e switch the order, since it is the
*REPLY: This will be done*

Supplementary material Data table: (1) a lot of blanks in the sample data, as well as an unknown words or sample core names below the data seat (see row 153-157).
*REPLY: These open fields in the spread sheet of the raw data will be marked 'below detection' in the revised version of the manuscript.*

(2) in "iGDGTs in peats" sheet, cite the references
*REPLY: This will be done*

(3) in "Lignite crenarchaeol", Sluijs et al. reported the GDGTs (iso- and br-) data originally from Naaf et al. (2018) and their newly measured 'Cren. Isomer'. Here, I suggest the authors to report the other iso and br-GDGTs abundances (here which I assume is HPLC/MS integrated peak area) together since they clearly mentioned in 'Material and Methods' that they re-analyzed the polar fraction of the lignite samples. Although I expect that this will not significantly change the result, still comparing only the newly measured 'Cren. Isomer' with reported GDGT dataset is not acceptable. This is because even measuring the same sample in the same method, the peak area can be different among interlaboratory measurements, the analytical parameter of the analytical instrument etc.
*REPLY: We did not re-analyse these samples, but instead revisited the original chromatograms where we determined the peak area for the crenarchaeol isomer (i.e., Naafs et al., 2018). We have amended the text to make this clearer.*

In addition, I suggest to add the calculations and results of the 'fraction of isoGDGTs' in all lignite samples. Lastly, minor comment on style of the table (e.g. missing cell borders, missing compound names) to be consistent. Describe 'n.d.' and 'b.d' too.
*REPLY: We will add steps in our calculated values for fraction of isoGDGTs in lignites to our data supplement. In addition, to further facilitate reproducibility, we add an example calculation of the data presented in Supplementary Figure 1 and include the meaning of the abbreviations.*

**References**

Naafs, B. D. A., McCormick, D., Inglis, G. N., and Pancost, R. D., 2018. Archaeal and bacterial H-GDGTs are abundant in peat and their relative abundance is positively correlated with temperature, *Geochim Cosmochim Ac*, 227, 156-170.

Sinninghe Damsté, J. S., Rijpstra, W. I. C., Hopmans, E. C., den Uijl, M. J., et al., 2018. The enigmatic structure of the crenarchaeol isomer, *Org Geochem*, 124, 22-28.

Sluijs, A., Schouten, S., Pagani, M., Woltering, M., et al., 2006. Subtropical Arctic Ocean temperatures during the Palaeocene/Eocene thermal maximum, *Nature*, 441, 610-613.

Sluijs, A., Brinkhuis, H., Crouch, E. M., John, C. M., et al., 2008a. Eustatic variations during the Paleocene-Eocene greenhouse world, *Paleoceanography*, 23, PA4216.

Sluijs, A., Röhl, U., Schouten, S., Brumsack, H.-J., et al., 2008b. Arctic late Paleocene–early Eocene paleoenvironments with special emphasis on the Paleocene-Eocene thermal maximum (Lomonosov Ridge, Integrated Ocean Drilling Program Expedition 302), *Paleoceanography*, 23, PA1S11.

Sluijs, A., Schouten, S., Donders, T. H., Schoon, P. L., et al., 2009. Warm and Wet Conditions in the Arctic Region during Eocene Thermal Maximum 2, *Nature Geoscience*, 2, 777-780.

Suan, G., Popescu, S.-M., Yoon, D., Baudin, F., et al., 2017. Subtropical climate conditions and mangrove growth in Arctic Siberia during the early Eocene, *Geology*, 45, 539-542.

Weijers, J. W. H., Lim, K. L. H., Aquilina, A., Sinninghe Damsté, J. S., et al., 2011. Biogeochemical controls on glycerol dialkyl glycerol tetraether lipid distributions in sediments characterized by diffusive methane flux, *Geochemistry, Geophysics, Geosystems*, 12.

---

## Referee Report (RR1)

In this work, Sluijs et al. revisit the early Paleogene sedimentary records recovered from the central Arctic Ocean (2004 ACEX Expedition) to better examine the past sea surface temperature (SST) estimates from their earlier studies (Sluijs et al., 2006; 2009). In their previous study which lead to major scientific discoveries of hyperthermal events and climatic/ecological changes in the Arctic region during the late Paleocene – early Eocene, the biomarker records revealed unusual GDGT (glycerol dialkyl glycerol tetraether) assemblage (i.e. high GDGT-3) and high terrestrial organic input, leaving some concerns in the application of GDGT-based paleothermometry in the Arctic (e.g. $TEX_{86}$). Here, the study re-evaluates the previous record/samples with new analyses, including glycerol monoalkyl glycerol tetraethers (GMGTs) for recently developed temperature proxy.

First of all, $TEX_{86}$ is a charming proxy that could help us learn the past ocean temperature, especially in the greenhouse climate. The study provides an excellent and thorough reassessment of the TEX86, especially describing the current status of understanding/limitation of the $TEX_{86}$-SST calibration and testing the potential bias from other GDGT sources/factors with the range of indices in their actual samples. In addition, the fractional calculation of terrestrial-derived isoprenoid GDGTs provides valuable insight into $TEX_{86}$ application and I appreciate the authors for adding the example calculation in the spreadsheet. $TEX_{86}$-H (Kim et al., 2010) and BAYSPAR (Tierney & Tingley, 2014) has been used after screening the bias and I also agree with the authors (in their Author's Response) for excluding the OPTiMAL (Eley et al., 2019) method which is yet subject to significant revisions.

Secondly, the study provides the several other independent temperature proxies (e.g. pollen, terrestrial vegetation indicators) along with the GDGT-base proxies and suggests that the SST estimate of ~20-25 °C is skewed toward the summer-end. Importantly, I find this study as an open question study on how the current paleothermometry (i.e. $TEX_{86}$, MBT) should be applied, what are the limitations, and what should be considered, especially when studying the Arctic region during past greenhouse climate condition. This study would be a broad interest to geochemists, paleoceanographers and climate modelers.

Lastly, I find the MS greatly improved after the first revision. The abstract, main text, the load of citations and figures all have become concise and clear. Here, I only have few minor comments which are mostly technical corrections (see below).

Overall, I recommend this work to be accepted in *Climates of the Past* for providing the next stepping stone of temperature reconstruction based on our current understanding of the proxy application and the importance of studying the nature of past Arctic Ocean, which is an critical end-member in assessing the past polar warmth and latitudinal temperature gradient of the Earth climate system for both proxy and climate model studies.

Some minor comments are below:

Line 223: I suggest to remove the word "assumed" since these diagnostic GDGT distributions (predominantly GDGT-1 to -3) have been observed in many studies from methane-impacted

environments (e.g. Pancost et al., 2001) and based on this rationale, MI was developed. Rather add "which preferentially produce GDGT-1, 2 and 3 (refs)";

Line 286: calculate from $TEX_{86}$ using;

Line 289: limits of the $TEX_{86}$-RI relationship;

Line 304: brGDGT-IIa';

Line 306: brGMGT-H1048;

Line 329: duplicate parenthesis;

Line 350: does this mean that the original BIT is slightly higher than the newly measured BIT? Then I assume "0.05" higher rather "0.5" which this will be a huge difference;

Line 398-399: interval 371-369 "also" stands out;

Line 416-417: "cyclization" same as line 416. Also is the degree of cyclization interpreted using RI (Figure 6e, red)?

Line 421: section 2.2.6;

Line 460: seen in GDGT-2 and;

Line 610: H-MBT (H1020c, H1034b);

Fig.3

Based on provided supplement data, I find that the two regression lines of TEX86 and BIT are both somewhat different from Figure 3.

To show this, I simply plotted the TEX86 relationship (new vs. original) from the data spreadsheet – column BQ (for newly measured TEX86 data, excluding five outliers mentioned in the author's response) and column DC (for original TEX86 data) – which gives regression of y=0.836x+0.077 (R^2=0.83; see figure below). Slightly lower slope than 1 indicates that the newly TEX86 is slightly higher the original TEX86, which is also shown in the TEX86 vs. depth plot (MS Figure 3 left; red generally higher than blue). For BIT, regression gives y=1.055x+0.021 (R^2=0.97) – column BR (new BIT) vs. column DB (original BIT).

Please verify if the regression curve needed to be corrected.

[Figure]

**Fig. 1.** TEX$_{86}$ relationship (new vs. original data) plotted from the supplementary data spreadsheet. Regression shown in red line and 1:1 line in gray.

---

## Author Response (AR2)

Dear editor,

We thank both of the reviewers for taking another good look at the manuscript and are happy to see both are supportive of publication. Below, we indicate how we made our final changes based on the remaining suggestions by reviewer #1. Reviewer #2 did not provide any comments.

In addition to these corrections, we have included the novel BayMBT calibration for the MBT'$_{5ME}$ proxy (Dearing Crampton-Flood et al., 2020) in addition to the calibration by De Jonge et al. (2014) in Figure 7, and briefly discussed the implication at the end of the proxy comparison section 5.3.2.

Sincerely, also on behalf of my co-authors,

Appy Sluijs

**Anonymous Reviewer #1**

Some minor comments are below:

Line 223: I suggest to remove the word "assumed" since these diagnostic GDGT distributions (predominantly GDGT-1 to -3) have been observed in many studies from methane-impacted environments (e.g. Pancost et al., 2001) and based on this rationale, MI was developed. Rather add "which preferentially produce GDGT-1, 2 and 3";
**Reply.** *We have specified this and added the Pancost et al. (2001) reference.*

Line 286: calculate from TEX86 using;
**Reply.** *We think 'calculated' is correct here.*

Line 289: limits of the TEX86-RI relationship;
**Reply.** *done.*

Line 304: brGDGT-IIa';
**Reply.** *corrected.*

Line 306: brGMGT-H1048;
**Reply.** *corrected.*

Line 329: duplicate parenthesis;
**Reply.** *corrected.*

Line 350: does this mean that the original BIT is slightly higher than the newly measured BIT? Then I assume "0.05" higher rather "0.5" which this will be a huge difference;
**Reply.** *We have deleted this confusing number; the different slope mentioned in the sentence is clear.*

Line 398-399: interval 371-369: "also: stands out;
**Reply.** *We chose to keep the 371.0 and 369.0 as was because the decimals matter. We included 'also'..*

Line 416-417: "cyclization" same as line 416. Also is the degree of cyclization interpreted using RI (Figure 6e, red)?
**Reply.** *We chose to retain both 'cyclization' to keep the sentence clear; we added 'e' to 'Figure 6e'.*

Line 421: section 2.2.6;
**Reply.** *done.*

Line 460: seen in GDGT-2 and;
**Reply.** *Changed to isoGDGT-2.*

Line 610: H-MBT (H1020c, H1034b);

*Reply.* *Corrected.*

Fig.3
Based on provided supplement data, I find that the two regression lines of TEX86 and BIT are both somewhat different from Figure 3.
To show this, I simply plotted the TEX86 relationship (new vs. original) from the data spreadsheet – column BQ (for newly measured TEX86 data, excluding five outliers mentioned in the author's response) and column DC (for original TEX86 data) – which gives regression of $y=0.836x+0.077$ ($R^2=0.83$; see figure below). Slightly lower slope than 1 indicates that the newly TEX86 is slightly higher the original TEX86, which is also shown in the TEX86 vs. depth plot (MS Figure 3 left; red generally higher than blue). For BIT, regression gives $y=1.055x+0.021$ ($R^2=0.97$) – _column BR (new BIT) vs. column DB (original BIT). Please verify if the regression curve needed to be corrected.

*Reply.* *We thank the reviewer for noticing; there was indeed a mistake in both the regressions, apparently due to a software error. We have now corrected this in Figure 3 and adapted the first paragraph of the results section accordingly.*